# LAYER-WISE KNOWLEDGE DISTILLATION FROM A PRETRAINED NETWORK IMPROVES HYPERNETWORK CONVERGENCE

## ABSTRACT

Hypernetworks that generate weights of another network often exhibit lower test accuracy and slower convergence due to implicit weight updates. The recently proposed HyperLight framework (Magnitude Invariant Parameterisations, MIP) addresses this convergence issue by bounding the scale of the hypernetwork's input encoding using sine-cosine transforms and by introducing additive weights. Preliminary experiments revealed that when deeper primary networks are fully *hypernetised*, MIP achieves lower test accuracy compared to a canonically trained network. This paper investigates layer-wise knowledge distillation methods for hypernetwork training by bridging the *hypernetised* layers with a pretrained *Teacher* network of the same architecture. Nine layer-wise KD methods (*FeatureKD*) – AB, AT, CwD, FitNets, FSP, FT, JacobianKD, RKD, and SP – were evaluated on the `shufflenetv2_0x5` architecture for the CIFAR-100 classification task. The two best-performing methods, AB-KD (Activation Boundary) and AT-KD (Attention Transfer), were further evaluated on nine additional deep networks, including ShuffleNet, ResNet, MobileNet, VGG, and Reparameterised VGG. Experiments reveal that AT and AB methods applied to MIP hypernetworks improve performance even for fully *hypernetised* deeper networks such as VGG19. For example, AB-KD with MIP achieved a test accuracy of $72.65\%$, only $1.22\%$ lower than the canonically trained *Teacher* at $73.87\%$, compared to the MIP baseline accuracy of $11\%$.

## 1 INTRODUCTION

Hypernetworks Ha et al. (2017), which generate weights of a primary network using a single backpropagation loop, facilitate a wide spectrum of applications: In one extreme, hypernetworks are applied for generalising weights of a primary network across multiple tasks or data distributions, such as in Bayesian Neural Networks, Multi-Task Learning, generative methods, adversarial defence, meta-learning, federated learning, and few-shot learning; in the other extreme of applications, hypernetworks improve a primary network on a single task or a data distribution, such as in compression methods including pruning and *chunk-by-chunk* generation of weights for larger models Ha et al. (2017). To benefit from hypernetworks in this wide spectrum of applications, the main hurdles in practice are the difficulty in hypernetwork convergence and the lower test accuracy of the hypernetwork-generated (*hypernetised*) primary network compared to its canonically trained counterpart.

Achieving hypernetwork convergence and improving the primary network's test accuracy are challenging due to the inherent complexity introduced by its implicit weight update mechanism. During the forward pass, a hypernetwork generates weights for the *hypernetised* layers of the primary network, followed by output prediction and loss computation. *hypernetised* parameters of the primary network act as placeholders without being updated directly, but implicitly via updates to the hypernetwork parameters. During the backward pass, a single backpropagation loop updates the hypernetwork parameters (and if any learnable parameters remain in the primary network without hypernetisation), resulting in rapidly changing activations and gradients.

Several attempts have been made for hypernetwork convergence mainly through weight initialisation, including orthogonal weight initialisation Ha et al. (2017), maintenance of constant variance across layers of the primary network Chang et al. (2023) in Hyperfan, and approximating variance of a canonically trained primary network for weights and a desired mean value (e.g., zero) for biases as in HyperInit Beck et al. (2023). Section 2 discusses the practical issues that arise with these initialisation methods. Recently proposed Magnitude Invariant Parameterisation (MIP), also known as *HyperLight*, analyses the issues of exploding activations and diminishing gradients due to their dependency with the scale of the input $\gamma$. MIP utilises additive weights and sine-cosine-based input encoding ($\gamma$) to constrain the magnitude of activations and gradients. Furthermore, rather than simply replacing the weights of the primary network that leads to rapid changes, MIP utilises additive weights with learnable parameters for initial weights. Using these two extensions, MIP produced better hypernetwork convergence and stability that was empirically evaluated on

three generalisation applications including Bayesian ConvNets, parameter tuning for medical image registration, and learning features for multiple scales for images. In this research, initial experiments conducted revealed that MIP also struggles to converge and reach the test accuracy of the canonically trained version when the primary network is deep and almost fully *hypernetised*, i.e. when all convolutional blocks of the primary network are generated using the hypernetwork. For instance, `vgg11_bn, vgg19_bn` produced $46.93\%, 11.00\%$ test precision on the classification of CIFAR100 when all convolutional layers are *hypernetised* after 100 epochs, despite the heavy computational cost; canonically trained versions from *TorchHub* Chen report $70.78\%, 73.87\%$, respectively.

The challenges of training a deeper, almost fully *hypernetised* primary network stem from several reasons. First, the gradient from the last layer, especially for primary networks without residual connections, decays for the early layers as well as to the hypernetwork layers. Second, multiple gradient vectors flow simultaneously via (virtually) parallel pathways from early to last layers of the primary network to update middle layers of the hypernetwork. This phenomenon is likely to cause a *tug-of-war* contributing to rapid changes in activations and gradients after gradient updates. This is similar to the conflict between gradient vectors in *Tragic Triads* Yu et al. (2020b) occurring in Multi-Task Learning, though in a different form for hypernetworks. To alleviate the first challenge, a simple solution would be to introduce a heuristic-based loss term for each layer output, producing shorter paths for the gradient to flow, but such a heuristic might hinder holistic convergence. On the other hand, a multitude of techniques have been investigated to distil knowledge Hinton et al. (2015); Gou et al. (2021) from a *Teacher* network to a *Student* network at intermediate layers - commonly known as *Feature-KD*, such as Fitnets Romero et al. (2014), Attention Transfer Zagoruyko & Komodakis (2017), FSP Yim et al. (2017). Since a canonically trained primary network exhibits better performance, defining layer-wise knowledge distillation loss terms based on layer activations and gradients for each *hypernetised* layer can provide shorter paths for the gradient to travel, and towards clearer directions without causing conflicts.

To handle the hypernetwork convergence issue, this paper investigates layer-wise knowledge distillation methods in a unified framework, termed *LayerKD-HN*, where a canonically pre-trained, yet architecturally equivalent *Teacher* network is bridged with *hypernetised* layers of the primary network, acting as the *Student*. We investigate nine (9) layer-wise knowledge distillation methods, namely (1) FitNets Romero et al. (2014), (2) Attention Transfer (AT) Zagoruyko & Komodakis (2017), (3) Activation Boundary (AB) Heo et al. (2019), (4) Relational KD (RKD) Park et al. (2019), (5) Channelwise Distillation (CwD) Shu et al. (2021), (6) Similarity Preservation (SP) Tung & Mori (2019), (7) Jacobian KD Srinivas & Fleuret (2018), (8) Factor Transfer (FT) Kim et al. (2018), (9) Flow of Structure Procedure (FSP) Yim et al. (2017) . The former eight (1-8) calculate KD loss per each layer's output activation maps compared with the corresponding layer of the *Teacher* each. FSP calculates a matrix for each pair of adjacent layers of a network, resulting in knowledge distillation between the *Student* and the *Teacher* using those matrices. These KD methods, some of which belong to instance-relations-based KD (e.g. RKD), can be transformed for layer-wise bridging with a *Teacher* as described in this paper. Further, the proposed method that approximates a single *Teacher* favours one extreme of hypernetwork applications on improving a primary network on a single task or a dataset, such as for pruning, over the other extreme of generalisation applications. Classic soft-logit based knowledge distillation at the final layer has been explored in Wu et al. (2023) and Xiong et al. (2024). To the best of the authors' knowledge, this is the first systematic study of layer-wise KD that incorporates multiple empirically successful distillation methods to directly improve hypernetwork convergence.

1. This paper investigates layer-wise knowledge distillation to improve hypernetwork convergence. In particular, a hypernetwork that uses Magnitude Invariant Parameterisations (MIP/HyperLight) was investigated for layer-wise knowledge distillation from a pretrained version of the primary network by bridging layers and incorporating layer-wise knowledge distillation loss terms.

2. Nine layer-wise knowledge distillation methods, namely AB, AT, CwD, FitNets, FSP, FT, JacobianKD, RKD, and SP, were empirically tested in the *HyperLight* framework in a unified manner on primary networks of type `ShuffleNetv2_0x5, ShuffleNetv2_1x5, MobileNetv2_0x5, MobileNetv2_1x4, ResNet20, ResNet56, VGG11_bn, VGG19_bn`, and (reparameterised) `repVGG11, repVGG19` on the CIFAR100 image classification dataset. Experimental results suggest that the Attention Transfer (AT) Type-0 and Activation Boundary (AB) methods improve the results of MIP-based hypernetworks when all 2D convolutional layers are *hypernetised*.

3. Empirically, uniform weights for layer-wise KD loss terms were found to perform better than dynamic weights that are constructed using a Gaussian spike moving from the early layers to the last layer with the epoch count.

## 2   RELATED WORK

**Hypernetwork Convergence** Hypernetworks were first introduced in Ha et al. (2017), in static (fully connected) and dynamic (RNN-based) forms, and were trained with specialized architectural modifications and dataset-specific

weight initialisations such as orthogonal weight initialization Ha et al. (2017). Chauhan et al. (2024) provides a recent review of hypernetworks in deep learning.

In canonical training, gradients update all parameters directly, whereas in hypernetwork training, updates apply only to the hypernetwork (and any non-hypernetised layers), which then generates the target parameters. Although hypernetworks enable a wide spectrum of applications, their convergence has largely relied on ad hoc, trial-and-error heuristics rather than principled approaches Chang et al. (2023).

The first principled method for hypernetwork convergence was *HyperFan* Chang et al. (2023), which generalised variance-based schemes such as fan-in and fan-out to hypernetworks, producing *Hyperfan-in* and *Hyperfan-out*. Hyperfan-in preserves activation variance in the primary network, while Hyperfan-out preserves gradient variance. Maintaining both simultaneously remains challenging. Beck et al. (2023) further noted the method's non-uniform behaviour across weights and biases and its strong dependence on the chosen activation function.

To address these limitations, Beck et al. (2023) proposed Weight-HyperInit and Bias-HyperInit for meta reinforcement learning. Weight-HyperInit approximates the variance of a standard Glorot or Kaiming-initialised primary network, while Bias-HyperInit constrains generated biases toward a desired mean (e.g., zero). For deeper networks with heterogeneous activation functions, however, maintaining consistent variance (as in Hyperfan) or approximating standard statistics (as in HyperInit) remains impractical.

Magnitude Invariant Parameterisation (MIP, also termed HyperLight) Ortiz et al. provided a more systematic approach by analysing and empirically demonstrating the dependence of activations and gradients on the scale of the hypernetwork input encoding, $\gamma$. Two key contributions were proposed: (i) bounding $\gamma$ via concatenated sine–cosine transforms to mitigate exploding activations and gradients, and (ii) introducing additive rather than replacement weights, stabilising training by allowing generated weights to augment rather than overwrite learnable base parameters. MIP was evaluated across tasks such as Bayesian ConvNets for classification, HyperMorph for medical image registration, and feature spatial scaling, each using different interpretations of $\gamma$ (e.g., priors, regularisation coefficients, or scale parameters). Despite its effectiveness on generalisation tasks, preliminary experiments showed that MIP struggles when all convolutional layers of a deeper ConvNet are hypernetised. This motivates our work, which builds on MIP's convergence strengths and investigates layerwise knowledge distillation for further improvement.

**Knowledge Distillation for Hypernetwork Applications** Knowledge distillation has been applied to hypernetworks primarily in the form of classic soft-logit KD Hinton et al. (2015). For instance, HyperINR Wu et al. (2023) incorporated soft-logit KD at the final layer, while HyperDistill Xiong et al. (2024) employed the same formulation for policy distillation, mapping task descriptors such as morphology and attributes into task-conditioned embeddings via a hypernetwork.

**Layer-wise Knowledge Distillation** Classic soft-logit KD was first extended to feature representations by FitNets Romero et al. (2014), followed by AT Zagoruyko & Komodakis (2017), which distils attention maps and gradients, and AB Heo et al. (2019), which transfers margin-based information in an SVM-like formulation. Other variants include RKD Park et al. (2019), CwD Shu et al. (2021), SP Tung & Mori (2019), Jacobian KD Srinivas & Fleuret (2018), FT Kim et al. (2018), and FSP Yim et al. (2017), each introducing distinct mechanisms for aligning feature-level knowledge. AT and AB are discussed in detail in Sections 3.3.1 and 3.3.2, with the remaining methods summarised in Section A.3.

## 3 IMPROVING HYPERNETWORK CONVERGENCE USING LAYERWISE KNOWLEDGE DISTILLATION

### 3.1 PROBLEM FORMULATION, OBJECTIVE, AND NOTATION

Given a dataset $\mathcal{X} \times \mathcal{Y} = \{(x, y) \mid x \in \mathcal{X} \subseteq \mathbb{R}^{w_{in} \times h_{in} \times c_{in}}, \ y \in \mathcal{Y}\}$ consisting of tensor inputs and corresponding class labels, a *hypernetised* primary network $f(\cdot)$ is trained to predict labels $\tilde{\dagger}$.

The primary network $f(\cdot)$ is partially or fully parameterised by a hypernetwork $h(\gamma; [W_1, \ldots, W_H])$, which *hypernetises* $m$ layers while leaving the remaining layers (if any) updated directly through backpropagation. For each *hypernetised* layer, parameters $\theta_m$ are generated by the hypernetwork $h(\gamma; [W_1, \ldots, W_H])$ (details in Section 3.2). During training, gradient updates are applied only to the hypernetwork parameters $[W_1, \ldots, W_H]$ and to the non-*hypernetised* layers of $f(\cdot)$. The parameters of the *hypernetised* layers are therefore not directly optimised via backpropagation but are instead learned indirectly through the hypernetwork.

Due to the added complexity of hypernetwork-based weight generation Ortiz et al., the *hypernetised* primary network often underperforms compared to a canonically trained network (*Teacher*) $g(\cdot)$ of identical architecture that is trained end-to-end by standard backpropagation. The objective of this work is thus to improve the joint training of $f(\cdot)$ and $h(\cdot)$, enhancing test accuracy of the *hypernetised* network so that it approaches the performance of the Teacher $g(\cdot)$. To this end, this work investigates **layerwise knowledge distillation** as a mechanism to transfer information from $g(\cdot)$ to the *hypernetised* $f(\cdot)$.

### 3.2 BACKGROUND ON MAGNITUDE INVARIANT PARAMETERISATION (MIP/HYPERLIGHT) HYPERNETWORKS

Magnitude Invariant Parametrisation (MIP) Ortiz et al. mitigates the exploding activations and gradients of hypernet-works through two key modifications: (1) bounding the input encoding $\gamma$ within $[0, 1]$ using sine–cosine or min–max transforms, and (2) learning additive weights to a set of base weights $\theta_0$, which are themselves trainable parameters.

Specifically, a scalar input $\gamma \in \mathbb{R}$ to the hypernetwork is transformed into $z_0 = \left[ \sin\left(\frac{\pi\gamma}{2}\right), \cos\left(\frac{\pi\gamma}{2}\right) \right]$. The hyper-network is implemented as a fully connected network $h(\gamma; \omega)$ with $H - 1$ hidden layers, where $H$ is set to $4$ in this work (see Figure 1). In our experiments, the hidden layer widths are either $[16, 64, 128]$ or $[16, 16, 16]$. The final layer produces a buffer that is reshaped into the *hypernetised* parameters of the primary network according to Eq. 3, where $p$ and $q$ denote the buffer indices assigned to layer $m$. Rather than replacing the primary network weights entirely with hypernetwork outputs, MIP generates additive updates that are applied to the learnable base parameters $\theta_0$. These base parameters continue to be refined through backpropagation, while the hypernetwork learns to generate residual adjustments, thereby improving stability and convergence.

$$z_k \leftarrow \phi(W^{(k)} z_{k-1} + b^{(k)}), \quad k = 1, ...(H - 1) \tag{1}$$

$$\Delta\theta_m \equiv h(\gamma; W) \leftarrow W^{(H)} z_{H-1} + b^{(H)} \tag{2}$$

$$\theta_m \leftarrow \text{reshape}(h(\gamma; W)[p : q] + \theta_0[p : q]) \tag{3}$$

Activation maps are represented as $f_m(x)$, $g_m(x)$ for a subnetwork from the input layer to layer $m$. Input encoding to the hypernetwork $\gamma \in \mathbb{R}$ can be generalised to a vector.

### 3.3 LAYERKD-HN: A UNIFIED FRAMEWORK FOR EVALUATING DIFFERENT LAYERWISE KNOWLEDGE DISTILLATION METHODS FOR HYPERNETWORK CONVERGENCE

A unified framework, denoted here as LayerKD-HN, was constructed to evaluate nine (9) different layer-wise KD methods in order to examine whether each method supports hypernetwork convergence, in particular Magnitude Invariant Parametrisation (MIP) Ortiz et al.. LayerKD-HN unifies layer positioning, weight assignment, and the order of training steps—that is, whether knowledge distillation is conducted prior to classification or simultaneously.

As shown in Figure 1, the activation maps $f_m(x_i; h(\gamma))$ from each *hypernetised* layer $m$ of the primary network are supported in the learning of the classification (main) task by adding layer-wise knowledge distillation loss terms, com-paring them with the activation maps $g_m(x_i; \zeta_0)$ from the same layer $m$ of a pretrained (*Teacher*) network. Different training schemes and layer choices were unified in the proposed framework, in contrast to their original formulations. For instance, FitNet-KD Romero et al. (2014) originally considers only a single middle layer, whereas this work applies the same loss across all convolutional layers, as detailed in the following sections and in the Appendix.

Sections 3.3.1 and 3.3.2 describe the Attention Transfer (AT) and Activation Boundary (AB) KD methods, respectively, due to space limitations. Appendix A.3 provides details of the remaining seven methods—Channel-wise Distillation (CwD) Shu et al. (2021), Flow of Solution Procedure (FSP) Yim et al. (2017), Factor Transfer (FT) Kim et al. (2018), FitNets-KD Romero et al. (2014), Jacobian-KD Srinivas & Fleuret (2018), Relational KD (RKD) Park et al. (2019), and Similarity Preservation (SP) Tung & Mori (2019)—together with their configurations.

### 3.3.1 BACKGROUND ON ACTIVATION TRANSFER (AT) KD

In Attention Transfer (AT) Zagoruyko & Komodakis (2017), spatial attention is defined by flattening the channel dimension of the feature maps from each channel $c$ of the subnetwork $f_m$ (or $g_m$) using a summary metric. This yields three subtypes of AT, denoted $S^{AT*,p}$, where $*$ indicates the variant: Type 0 – sum of absolute values (Eq. 4); Type 1 – sum of absolute values raised to the power $p > 1$ (Eq. 5); Type 2 – maximum of absolute values raised to the power



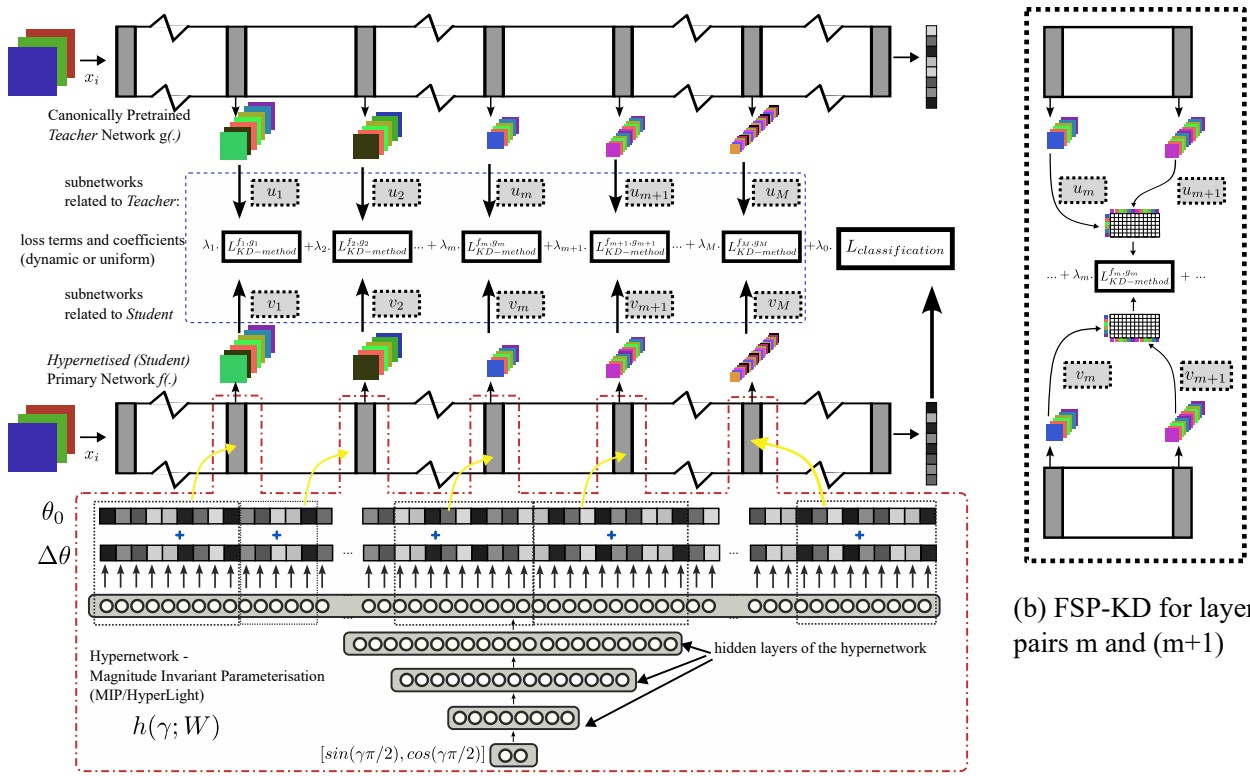

(b) FSP-KD for layer pairs m and (m+1)

(a) Unified framework for KD methods: AB, AT, CwD, FitNet-KD, FT, JacobianKD RKD, SP

Figure 1: An abstract representation of LayerKD-HN, which unifies and evaluates layer-wise knowledge distillation methods for hypernetwork convergence. The methods include AB, AT, CwD, FSP, FT, FitNets-KD, Jacobian-KD, RKD, and SP. A Magnitude Invariant Parametrisation (MIP)-based hypernetwork receives an input $\gamma$ and produces additive weights for the *hypernetised* layers of the primary network (*Student*). A pretrained *Teacher* with equivalent architecture to the primary network generates activation maps that are used for the layer-wise knowledge distillation loss terms (Sections 3.3.1, 3.3.2, A.3). Subfigure (b) illustrates the key difference of FSP-KD, which utilises pairs of *hypernetised* adjacent layers to produce the matrices $S^{f_m, f_{(m+1)}}$ and $S^{g_m, g_{(m+1)}}$ (Section A.3.7).

$p \geq 1$ (Eq. 6).

$$\text{AT Type 0:} \qquad S^{AT0,p}(f_m(x_i)) \triangleq \sum_{c \in C_m} |f_m^c(x_i; h(\gamma))| \tag{4}$$

$$\text{AT Type 1:} \qquad S^{AT1,p}(f_m(x_i)) \triangleq \sum_{c \in C_m} |f_m^c(x_i; h(\gamma))|^p \tag{5}$$

$$\text{AT Type 2:} \qquad S^{AT2,p}(f_m(x_i)) \triangleq \max_{c \in C_m} |f_m^c(x_i; h(\gamma))|^p \tag{6}$$

AT-KD loss is calculated by comparing the spatial attention of the feature maps for a batch $\mathcal{X}_B$ as Eq. 7.

$$\mathcal{L}_{AT-KD,p} = \sum_{m=1}^{M} \lambda_m \cdot \sum_{x_i \in \mathcal{X}_\mathcal{B}} \left\| \frac{vec(S_m^{AT*,p}(f_m(x_i; h(\gamma))))}{\|vec(S_m^{AT*,p}(f_m(x_i; h(\gamma))))\|_2} - \frac{vec(S_m^{AT*,p}(g_m^c(x_i; \zeta_0)))}{\|vec(S_m^{AT*,p}(g_m^c(x_i; \zeta_0)))\|_2} \right\|_p \tag{7}$$

### 3.3.2 BACKGROUND ON ACTIVATION BOUNDARY (AB) KD

Activation Boundary KD (Heo et al. (2019)) highlights the importance of transferring binary state of activations of neurons (or filters) between *Teacher* and *Student* despite the non-differentiability of the discrete mask function $\rho$ in Eq. 8 for calculating a mask-based KD loss in Eq. 9.

$$\rho(g_m^c(x_i)[a,b]) = \begin{cases} 1 & \text{if} \\ 0 & \text{otherwise} \end{cases} \tag{8}$$

$$\mathcal{L}_{mask} = \|\rho(f_m(x_i; h(\gamma))) - \rho(g_m(x_i; \zeta_0))\| \tag{9}$$

where $a, b$ representing spatial coordinates, $c$ representing the filter/channel. Resembling the SVM formulation, AB-KD utilises hinge loss to circumvent this by introducing a margin $\mu$ closer to the activation boundary, multiplied by the mask function as in Eq. 10.

$$\mathcal{L}_{AB-KD} = \sum_{m=1}^{M} \lambda_m . \sum_{x_i \in \mathcal{X}_B} \|\rho(g_m(x_i; \zeta_0)) \odot \sigma(\mu\mathbf{1} - f_m(x_i; h(\gamma)) ) \tag{10}$$
$$+ [1 - \rho(g_m(x_i; \zeta_0))] \odot \sigma(\mu\mathbf{1} + f_m(x_i; h(\gamma)) )\|_2^2$$

where $\odot$ representing element-wise product between vectors and $\mathbf{1}$ a vector of length of number of channels $|C_m|$ at layer $m$ of the networks. $\mathcal{L}_{AB-KD}$ penalises the wrong activations and maintains the margin $\mu$ from the activation boundary in the primary network $f_m$.

### 3.4 TRAINING PROCEDURE

Algorithm 1 presents a generalised pseudocode for evaluating each of the nine layerwise knowledge distillation methods in a uniform manner. Feature maps are extracted at each *hypernetised* layer $m$ of the primary network from $f_m$ and from the pretrained *Teacher* network $g_m$. The knowledge distillation loss term $\mathcal{L}_{KD-method}(f_m(\mathcal{X}_B; h(\gamma)), g_m(\mathcal{X}_B; \zeta_0))$ for each layer $m$ is calculated according to Equations 11–29, as indicated in Algorithm 1. An important distinction is that the FSP-KD method utilises pairs of layers for building the FSP matrices $S^{f_m, f(m+1)}$ and $S^{g_m, g(m+1)}$ and for calculating $\mathcal{L}FSP$ in Eq. 29. For each epoch, the $\lambda_m$ values are obtained as described in Section A.4. An optimiser $o_{[f+h]}$ updates the gradients for the hypernetwork parameters $W^{(k)}$ for $k = 1, \ldots, H$, as well as for the non-*hypernetised* parameters of the primary network requiring gradient calculations. Without loss of generality, all convolutional blocks were *hypernetised* in this paper using the `hyperlight.find_modules_of_type($f(\cdot)$, [torch.nn.Conv2d])` function, to maximally *hypernetise* nearly all layers of the hypernetwork. Note that classifier layers of some primary networks use Conv2D layers, which are also *hypernetised* using this function, whereas classifier layers using fully connected layers are not *hypernetised*.

For the Factor Transfer (FT-KD) method, a pretraining step with a reconstruction loss is required for the translator $u_m$ and paraphraser $v_m$ subnetworks described in Section A.3.2 for each *hypernetised* layer $m = 1, \ldots, M$. The number of pretraining epochs $N_{FT}$ is set to 20 in the experiments. Separate optimisers $o_m$ for $m = 1, \ldots, M$ and separate schedulers were utilised, with hyperparameters set to *mode = minimum*, `patience = 3`, and `factor = 0.5`. Paraphraser learning rates were adjusted empirically to avoid numerical instability, and Figure 7 in the Appendix depicts the reconstruction losses for each *hypernetised* layer of `ShuffleNetV2_0x5` on CIFAR100. For layers `conv5.0` and `conv5.1`, the learning rates of the corresponding Adam optimisers were set to 0.0001, while all optimisers corresponding to other layers were set to 0.001.

## 4 EXPERIMENTAL RESULTS

In this section, experiments conducted to investigate the following are presented.
1. Can the hypernetwork convergence be improved by using layer-wise knowledge distillation (LayerKD) applied to convolutional neural networks, in particular, reducing the number of epochs for convergence and improving test-set accuracy similar to a canonically trained network?
2. Can altering the KD coefficients $\lambda_m$ to emphasise layers from the first to the last parts of the primary network compared to uniform $\lambda_m$ assignments contribute to better performance?

### 4.1 EXPERIMENTS ON HYPERNETWORK CONVERGENCE USING LAYERWISE KD

This section presents experiments conducted for nine (9) layer-wise knowledge distillation methods for `ShuffleNet_x0_5` conducted on the CIFAR100 dataset. As *Teacher*, pretrained weights from Chen were used for knowledge distillation. Due to computation time, the best performing layer-wise KD method, namely, Attention Transfer (AT-KD) – Type 0, method and the baseline, were applied for the other nine network configurations on network architecture types: VGG, Reparameterised VGG, ResNet, MobileNet, and ShuffleNet, considering one shallow configuration and one deeper configuration for each. Memory utilisation of MIP heavily depends on the number of nodes in its penultimate hidden layer (last hidden layer's dimensionality), hence it had to be scaled from 128 to 16, so that the hidden layers had to be downscaled to [16, 16, 16] from [16, 64, 128] to facilitate execution on the server.

Table 1: Nine layer-wise knowledge distillation methods applied to evaluate the test set accuracy (%) for CIFAR100 Image Classification when using MIP Ortiz et al. to hypernetise all Conv2D blocks of shufflenetv2_0x5. Column headers indicate whether uniform or dynamic $\lambda_m$ (Appendix A.4) for layer-wise KD loss terms and the hidden layer sizes of the MIP hypernetwork. Convergence plots depicted, in Figure 3 for column 2 on Dynamic $\lambda_m$ for [16, 64, 128] hidden sizes of hypernet, in Figure 5 for column 3 for Uniform $\lambda_m$ for [16, 64, 128] hidden sizes of hypernet.

| Method | Dyn. $\lambda_m$, hid=[16, 64, 128] | uni.$\lambda_m$, hid=[16, 64, 128] | uni. $\lambda_m$ hid=[16, 16, 16] |
|---|---|---|---|
| *Teacher* | *67.82* | *67.82* | *67.82* |
| No-KD (baseline-MIP) | 62.26 [$\Delta$=-5.56] | 62.26 [$\Delta$=-5.56] | 62.36 [$\Delta$=-5.46] |
| AB-KD | 64.65 (+2.39 ↑) [$\Delta$=-3.17] | 65.57 (+3.31 ↑) [$\Delta$=-2.25] | 65.95 (+3.59↑) [$\Delta$=-1.87] |
| AT-0(p=1) | *66.89 (+4.63 ↑) [$\Delta$=-0.93]* | **67.04 (+4.78 ↑) [$\Delta$=-0.78]** | 66.38 (+4.02↑) [$\Delta$=-1.44] |
| AT-1(p=2) | 62.20 (-0.06 ↓) [$\Delta$=-5.62] | 61.58 (-0.68 ↓) [$\Delta$=-6.24] | 61.54 (-0.82↓) [$\Delta$=-6.28] |
| AT-1(p=4) | 62.20 (-0.06 ↓) [$\Delta$=-5.62] | 64.95 (+2.69 ↑) [$\Delta$=-2.87] | 65.11 (+2.75↑) [$\Delta$=-2.71] |
| AT-2(p=2) | 62.15 (-0.11 ↓) [$\Delta$=-5.67] | 62.67 (+0.41 ↑) [$\Delta$=-5.15] | 62.73 (+0.37↑) [$\Delta$=-5.09] |
| AT-2(p=4) | 63.16 (+0.90 ↑) [$\Delta$=-4.66] | 64.47 (+2.21 ↑) [$\Delta$=-3.35] | 64.58 (+2.22↑) [$\Delta$=-3.24] |
| CwD | 56.32 (-5.94 ↓) [$\Delta$=-11.50] | 55.35 (-6.91 ↓) [$\Delta$=-12.47] | 53.36 (-9.00↓) [$\Delta$=-14.46] |
| FSP | 60.49 (-1.77 ↓) [$\Delta$=-7.33] | 60.44 (-1.82 ↓) [$\Delta$=-7.38] | 59.45 (-2.91↓) [$\Delta$=-8.37] |
| FT | 61.50 (-0.76 ↓) [$\Delta$=-6.32] | 60.79 (-1.47 ↓) [$\Delta$=-7.03] | 61.85 (-0.51↓) [$\Delta$=-5.97] |
| FitNet-KD | 55.22 (-7.04 ↓) [$\Delta$=-12.60] | 53.34 (-8.92 ↓) [$\Delta$=-14.48] | 53.86 (-8.50↓) [$\Delta$=-13.96] |
| JacobianKD | 61.76 (-0.50 ↓) [$\Delta$=-6.06] | 61.89 (-0.37 ↓) [$\Delta$=-5.93] | 62.54 (0.18↑) [$\Delta$=-5.28] |
| RKD-A | 62.73 (+0.47 ↑) [$\Delta$=-5.09] | 62.25 (-0.01 ↓) [$\Delta$=-5.57] | 62.48 (+0.12↑) [$\Delta$=-5.34] |
| RKD-D | 62.72 (+0.46 ↑) [$\Delta$=-5.10] | 61.97 (-0.29 ↓) [$\Delta$=-5.85] | 62.07 (-0.29↓) [$\Delta$=-5.75] |
| RKD-DA | 62.17 (-0.09 ↓) [$\Delta$=-5.65] | 61.16 (-1.10 ↓) [$\Delta$=-6.66] | 61.85 (-0.51↓) [$\Delta$=-5.97] |
| SP | 61.97 (-0.29 ↓) [$\Delta$=-5.85] | 61.30 (-0.96 ↓) [$\Delta$=-6.52] | 62.05 (-0.31↓) [$\Delta$=-5.77] |

Table 2: Activation Boundary (AB) Heo et al. (2019) and Attention Transfer (AT-Type0) Zagoruyko & Komodakis (2017) improves the Test set accuracy (%) for CIFAR100 Image Classification when using MIP Ortiz et al. to hypernetise all Conv2D blocks of multiple deep ConvNets. Dynamic $\lambda_m$ (Appendix A.4) with a hypernetwork with [16,64,128] hidden sizes was utilised. Convergence plots depicted in Figure 2.

| Network | *Teacher* | No-KD (Baseline-MIP) | AB-KD | AT - Type 0 |
|---|---|---|---|---|
| shufflenetv2_x0_5 | *67.82* | 62.26 | 64.65 (+2.39 ↑) | **66.89 (+4.63 ↑)** |
| shufflenetv2_x1_5 | *74.23* | 66.93 | 69.34 (+2.41 ↑) | **70.95 (+4.02 ↑)** |
| resnet20 | *68.83* | 61.01 | **65.96 (+4.95 ↑)** | 64.76 (+3.75 ↑) |
| resnet56 | *72.63* | 64.34 | 65.73 (+1.39 ↑) | **67.96 (+3.62 ↑)** |
| mobilenetv2_x0_5 | *71.17* | 62.76 | 65.32 (+2.56 ↑) | **66.69 (+3.93 ↑)** |
| mobilenetv2_x1_4 | *76.29* | 62.16 | **70.73 (+8.57 ↑)** | 70.68 (+8.52 ↑) |
| vgg11_bn | *70.78* | 46.93 | **68.70 (+21.77 ↑)** | 61.99 (+15.06 ↑) |
| vgg19_bn | *73.87* | 11.00 | **72.65 (+61.65 ↑)** | 42.21 (+31.21 ↑) |
| repvgg_a0 | *75.30* | 69.09 | 63.11 (-5.98 ↓) | **71.05 (+1.96 ↑)** |
| repvgg_a2 | *77.50* | 67.98 | 59.63 (-8.35 ↓) | **72.20 (+4.22 ↑)** |

Figure 3 depicts the test accuracy with the use of a validation split with model selection: if the validation set's accuracy is improved after an epoch of training, the model is saved and the test set result is saved. Plots in Figure 3 contain empty epochs because the validation set accuracy was not improved in some of the latter epochs of 40 in total.

Table 1 shows the test accuracy for CIFAR100 classification with model selection using a validation set when using shufflenetv2_0x5 as the primary network and an MIP (HyperLight) hypernetwork Ortiz et al.. The training procedure was conducted for 40 epochs. The first column represents the dynamic $\lambda_m$ with parameters described in Section A.4 when the hidden dimensions of $h(.)$ are $[16, 64, 128]$. The second column contains the results for uniform $\lambda_m$ when the hidden dimensions of $h(.)$ are $[16, 64, 128]$. The third column represents uniform $\lambda_m$ when the hidden dimensions of $h(.)$ are $[16, 16, 16]$. The first row indicates the test accuracy of the canonically trained *Teacher*. Pretrained networks for the *Teacher* were downloaded from the TorchHub repository Chen. Overall, the best results are for uniform $\lambda_m$ with hidden dimensions for the middle layers of the hypernetwork $h(.)$ set to $[16, 64, 128]$.

Multiple pathways exist when a primary network is connected as shown in Figure 1 where middle layers of the hypernetwork is likely to undergo contrasting directions and magnitudes of gradient receiving from the early layers and the latter layers as shown for Multi-task Learning in Yu et al. (2020a). CIFAR100 classification results for

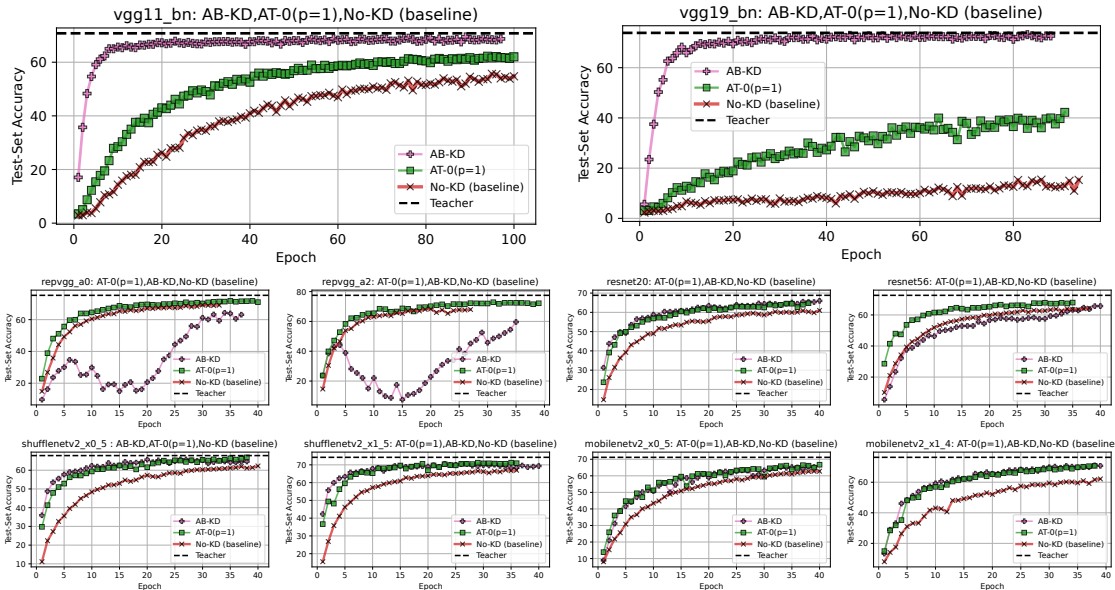

Figure 2: CIFAR 100 training for AB-KD, AT-0(p=1), and baseline (MIP without KD) for 10 models tested.

`ShuffleNetV2_0x5` to compare between the two approaches of (1) uniform $\lambda_m$ values versus (2) the dynamic $\lambda_m$ described in Section A.4 are presented in second and third columns of 1, respectively.

As shown in the first row of Table 1, baseline MIP without KD (No-KD) produced $62.26\%$ test accuracy after 40 epochs, compared to the canonically trained *Teacher* at $67.82\%$, with a gap of $5.56\%$. AT-type 0, denoted by AT-0(p=1), applied with uniform $\lambda_m$, performs best among all different configurations of the nine layer-wise KD methods added to MIP, with an improvement of $6.26\%$. Compared to the canonically trained version, the best performer is only $0.78\%$ behind. When AT-0 uses dynamic $\lambda_m$, MIP achieves an improvement of $4.63\%$ at $66.89\%$, just $0.93\%$ lower than the canonically trained *Teacher*. The lower-dimensional [16,16,16] hidden configuration in column 3 also produces $66.38\%$, which improves MIP by $4.02\%$. AT-type 2(p=4) also improves MIP by a small margin of $0.90\%$, while other AT configurations underperform compared to the No-KD baseline. The Activation Boundary (AB) KD method also improves MIP, with $2.39\%$ at $64.65\%$, which is the second-best result for `shufflenetv2_0x5` among the nine KD methods.

For dynamic $\lambda_m$ in column 2 of Table 1, AB-KD converges around the 25th epoch as shown in Figure 3 (c). The RKD-A and RKD-D methods slightly improve MIP by less than $0.5\%$, at $62.73\%$ and $62.72\%$, respectively, although their combination (0.5 each), termed RKD-DA, underperforms compared to the baseline. The rest of the KD methods perform worse than the baseline (No-KD). In particular, FitNet-KD, which was originally proposed for a single middle layer and is utilised here for knowledge distillation across all *hypernetised* layers, performs the worst, with a reduction of $7.04\%$ at $55.22\%$. Channel-wise Distillation (CwD) also produced $5.94\%$ lower test accuracy, at $56.32\%$, compared to the No-KD baseline. The Flow of Solution Procedure (FSP-KD), which uses pairs of layers per network to construct FSP matrices $S^{f_m, f_{(m+1)}}, S^{g_m, g_{(m+1)}}$ as in Eq. 28, also underperforms, with an accuracy of $60.49\%$. Factor Transfer (FT), which requires pretraining translators $u_m$ and paraphrasers $v_m$ for 20 epochs with a reconstruction loss, also underperforms compared to the baseline No-KD. Gradient-based Jacobian-KD with MIP not only consumes more training time due to complex computations, but also underperforms at $61.97\%$ compared to the baseline No-KD.

In summary, for CIFAR100 test accuracies with `shufflenetv2_0x5`, Attention Transfer-Type 0 (*AT-0 (p=1)*) and Activation Boundary (*AB-KD*) perform best and are therefore further evaluated on other networks described below.

### 4.1.1 AB-KD AND AT-0 ON MOBILENETS, SHUFFLENETS, AND RESNETS

Six networks of ShuffleNet, MobileNet, ResNet architectures from high and low computational demands as the primary network being *hypernetised* using MIP were tested with CIFAR100 classification task for 40 epochs as presented in Table 2. AB-KD and AT Type 0 (AT-0) both improve test accuracy of the baseline MIP for all the six networks namely `ShuffleNetv2_0x5`, `ShuffleNetv2_1x5`, `resnet20`, `resnet56`, `mobilenetv2_0x5`, and `mobilenetv2_x1_4`. The summary of the results on three versions AB-KD, AT-0, and No-KD in 40 epochs is as fol-

lows: AB-KD has the highest test accuracy among the above three variants for `resnet20, mobilenetv2_x1_4` at $65.96\%, 70.73\%$, which are improvements to baseline MIP (No-KD) by $4.95\%, 8.57\%$ , and with a gap from the canonically trained *Teacher* by $3.87\%, 5.56\%$, respectively. AT Type 0 (AT-0 (p=1)) has the highest test accuracy among the above three variants for `ShuffleNetv2_0x5, ShuffleNetv2_1x5, mobilenetv2_0x5, resnet56` at $66.89\%, 70.95\%, 66.69\%, 67.96\%$, which are improvements to the baseline MIP (No-KD) by $4.63\%, 4.02\%, 3.93\%, 3.62\%$ and with a gap from the canonically trained *Teacher* by $0.93\%, 3.28\%, 4.48\%, 4.67\%$, respectively.

### 4.1.2    AB-KD and AT-0 on VGGish Models with and without Residual Connections

*AB-KD* improves computationally more expensive models like VGG19 with batch normalisation (`VGG19bn`) by a large margin of $61.65\%$ as shown in Table 2 while the baseline MIP merely produced $11.00\%$ even after 100 epochs. The pretrained (*Teacher*) from TorchHub has test accuracy of $73.87\%$ and AB-KD performs very close to this, at $72.65\%$ which is merely a gao of $1.22\%$. AB-KD converges relatively very quickly compared to others within around 20 epochs as depicted in Figure 2 (b). AT-Type0 (AT-0 (p=1)) also improves the MIP baseline at $42.21\%$, with an increment of $31.21\%$. Important to note that VGG models can be difficult to train due to their lack of residual connections, yet the AB and AT-0 methods produce better convergence rates and test accuracies. AB-KD improves `VGG11bn` by $21.77\%$ at $68.70\%$. Compared to the (pretrained) *Teacher*'s test accuracy at $70.78\%$, AB-KD gets very close with a gap of $2.08\%$. This is achieved relatively quickly around $40^{th}$ epoch. Non-KD baseline severely struggled with $46.93\%$ even after 100 epochs, with a gap of $68.70\%$ from the canonically trained (*Teacher*) version of the primary network.

Reparameterised VGG modesl Ding et al. (2021) denoted by `repvgg_a0` and `repvgg_a2` which contains residual connections were improved with AT-0 closer to the canonically trained *Teacher*, even though AB-KD produced worse test accuracy for those two models, despite high performance in VGG. For `repvgg_a0`, AT Type 0 (AT-0(p=1)) converges within 40 epochs to $71.05\%$ improving the baseline MIP of $69.09\%$ with a gap of $1.96\%$ . Compared to the canonically trained *Teacher* at $75.30\%$, AT-0 has a gap of $4.25\%$ in 40 epochs, while baseline underperforming with a gap of $6.21\%$. For `repvgg_a2`, AT-0 produces $72.20\%$ compared to the MIP baseline at $67.98\%$ with a gap of $4.22\%$. Compared to the canonically trained *Teacher* from TorchHub, this is $5.3\%$ closer whereas the baseline was $8.52\%$. For both the `repvgg_a0` and `repvgg_a2`, AB-KD struggled in the middle epochs as in Figure 2 (c) and (d), and underperformed after 40 epochs at $63.11\%$ and $59.63\%$, respectively.

### 4.2    Discussion

The results presented in Section 4.1 indicate that the MIP baseline can be improved by adding layer-wise knowledge losses, in particular, either Attention Transfer Type 0 with p=1 (AT-0) or Activation Boundary (AB-KD) loss terms, even for deeper, expensive models such as `VGG19, ShuffleNetv2_1x5, MobileNetv2_1x4`. MIP was originally tested for scale-invariant training of MNIST where input encoding $\gamma$ acted as the scale value, although it was not properly tested for the other end of the applications such as for compression using Hypernetworks. Memory usage increases to higher values when using a hypernetwork based on the number of nodes in the penultimate layer of the hypernetwork, which is either 128 or 16 in the presented result. In future work, even though the presented method currently uses high memory requirements during training to host the hypernetwork of several magnitudes, the primary network can exhibit higher accuracies by using the proposed method for pruning tasks such as Li et al. (2020); Kumar et al. (2023).

## 5    Conclusion

The recently proposed Magnitude Invariant Parameterisation (MIP) alleviates hypernetwork convergence issues, yet fully *hypernetised* primary networks continue to underperform compared to canonical training. To address this gap, layerwise knowledge distillation (KD) is introduced, bridging each *hypernetised* layer with a pretrained *Teacher* network. Nine feature-level KD methods—AB, AT, CwD, FitNets, FSP, FT, JacobianKD, RKD, and SP—were systematically evaluated on ten architectures, including ShuffleNet, MobileNet, ResNet, VGG, and Reparameterised VGG. Results show that methods such as Attention Transfer (AT) and Activation Boundary (AB-KD) substantially enhance convergence, even for deeper networks like VGG19. Future work will explore extending the proposed *LayerKD-HN* framework to compression and other applications.

ACKNOWLEDGMENTS

GenAI was used in the following ways: (1)- (a) `www.chatgpt.com` was used to polish the writing after completing the authentic work of the authors, (b) for debugging purposes in implementation, and (c) to write Python codes for creating plots to be inserted in the paper (this is not for implementation of experiments). (2)- The Auto-correcting GenAI tool for grammar and improving writing available in `www.overleaf.com` was utilised.

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

## A APPENDIX

### A.1 EXPERIMENTAL SETUP

Experiments were carried out in PyTorch with HyperLight Ortiz et al. (2023); Ortiz et al. used to hypernetise primary networks. The experiments used a NVIDIA Tesla T4 with approximately 15GB CUDA memory and CUDA 12.7. The batch size was set to 16 to facilitate training larger models such as VGG models since hypernetisation increases the size of the combination depending on the dimensioality of the penultimate layer of the hypernetwork. The CIFAR100 data set was divided into 45000 for training, 10,000 for testing and 5000 for validation. The validation set is used for the selection of better performing weights. Implementations from Hu, and original implementations, and IrfanICMLL were referenced and transformed.

### A.2 TRAINING ALGORITHM FOR *LayerKD-HN*

### A.3 BACKGROUND ON LAYERWISE KNOLWEDGE DISTILLATION METHODS

#### A.3.1 BACKGROUND ON FITNETS KD

Fitnet loss is calculated as Eq.11 for each sub-network defined by $f_m$ and $g_m$ at layer m of the primary network and *Teacher* with equivalent archiecture with pretrained weights $\zeta_0$. Flattening a feature map to a vector is indicated by $vec(.)$. $C_m$ is set of filters available at layer $m$ and the activation map is indicated by $f_m^c, g_m^c$.

$$\mathcal{L}_{FitNets-KD} = \sum_{x_i \in \mathcal{X}_B} \sum_{m=1}^{M} \lambda_m \cdot \sum_{c \in C_m} ||vec(f_m^c(x_i; h(\gamma))) - vec(g_m^c(x_i; \zeta_0))|| \qquad (11)$$

---

**Algorithm 1:** Hypernetwork Training with Layerwise Knowledge Distillation

---

**Input:** Classification dataset $(\mathcal{X}, \mathcal{Y})$, Hypernetwork input encoding $\gamma$, MIP Hypernetwork $h(\gamma; \theta)$, Primary network $f(.)$ with $M$ layers to be *hypernetised* using MIP hypernetwork $h(\gamma)$, Pretrained *Teacher* network $g(.; h(\gamma))$ with pretrained weights $\zeta_0$, Layerwise Knowledge Distillation method, its hyperparameters, and auxiliary networks $u_m, v_m$ depending on the KD method

**if** *FT-KD* **then**
  Train paraphraser $u_m$ and translator $v_m$ networks using separate optimisers $o_m$ for unsupervised
  reconstruction loss in Eq. 12 for $N_{FT}$ epochs.

**for** *epoch* $n = 1, ..., N$ **do**
  **for** *batch* $\mathcal{X}_B \in \mathcal{X}$ **do**
   $\tilde{\mathcal{Y}}_B \leftarrow f(\mathcal{X}_B; h(\gamma))$
   Calculate classification loss $\mathcal{L}_{CE}(f(\mathcal{X}_B); \mathcal{Y}_B)$
   **if** *FSP-KD* **then**
    **for** *each layer* $m = 1, ..., M - 1$ **do**
     Calculate two matrices $S^{f_m, f_{(m+1)}}, S^{g_m, g_{(m+1)}}$ using adjacent pairs of *hypernetised* layer indices
     $m$ and $m+1$ as in Eq. 28.
     Use Eq. 29 with the two matrices $S^{f_m, f_{(m+1)}}, S^{g_m, g_{(m+1)}}$ to calculate $\mathcal{L}_{FSP}$ as Eq. 29.
   **else**
    **for** *each layer* $m = 1, ..., M$ **do**
     Calculate feature maps $f_m(\mathcal{X}_B; h(\gamma)), g_m(\mathcal{X}_B; \zeta_0))$
     **if** *Jacobian-KD* **then**
      Calculate gradients as input to Eq. 19 for JacobianKD
     Calculate $\mathcal{L}_{\text{KD-method}}(f_m(\mathcal{X}_B; h(\gamma)), g_m(\mathcal{X}_B; \zeta_0)))$ using relevant equation(s) for KD methods
     other than FSP 11.
   Calculate $\lambda_0$ for classification loss and $\lambda_m, m = 1, ..., M$ based on epoch $n$ as in Section A.4. *FSP-KD*
   requires $M - 1$ number of terms only.
   Using optimiser $o_{[f+h]}$, apply gradient step to hypernetwork parameters $W^{(k)}$ for $k = 1, ..., H$ and
   non-*hypernetised* parameters of $f$ calculated for loss $\mathcal{L}_{total} = \mathcal{L}_{CE} + \mathcal{L}_{\text{KD-method}}$

**Output:** $f(.; h(\gamma)), h(\gamma)$

---

### A.3.2 BACKGROUND ON FACTOR TRANSFER (FT) KD

Factor Transfer Kim et al. (2018) introduces two two auxiliary networks, namely *paraphraser* and *translator*. *Paraphraser* $u_m \equiv u_m^{enc} \circ u_m^{dec}$, an autoencoder with increasing channels in the middle layer without changing the spatial dimensions of feature maps, converts the *Teacher* factors to an "easier-to-learn" version. *Paraphraser* is trained with reconstructoin loss for the feature map outputs of the *Teacher* as in Eq. 12.

$$\mathcal{L}_{recon.}^{u_m} = \| g_m(x_i; \zeta_0) - u_m^{dec} \circ u_m^{enc} \circ g(x_i; \zeta_0)) \| \tag{12}$$

The intuition behind this unsupervised training is that *paraphraser* learns to "narrate" a lesson back to the *Teacher* with expanded details which becomes *Teacher factors* for transferring that as knowledge to the *Student*. In the experiments, this *paraphraser* training in Eq. 12 is conducted as a pretraining step before starting the training process which includes hypernetwork and primary network training supported by knowledge distillation (FT).

*Translator* $v_m$, an auxiliary convolutional neural network which outputs *Student* factors is trained to mimic *Teacher* factors from the paraphraser. The intuition behind having a separate *translator* sub-network is facilitating the flexibility to the *Student* to transform *Teacher* factors to its network.

Factor Transfer (FT) loss is defined in Eq. 13.

$$\mathcal{L}_{FT-KD} = \sum_{m=1}^{M} \lambda_m \cdot \left\| \frac{u_m^{enc} \circ g_m(x_i; \zeta_0)}{\| u_m^{enc} \circ g_m(x_i; \zeta_0) \|_2} - \frac{v_m \circ f_m(x_i; h(\gamma))}{\| v_m \circ f_m(x_i; h(\gamma)) \|_2} \right\|_p \tag{13}$$

In the experiments, p is set to 1.

### A.3.3 BACKGROUND ON SIMILARITY PRESERVATION (SP) KD

Similarity Preservation (SP) (Tung & Mori (2019)) distils knowledge between (pretrained) *Teacher* $g_m(.; \zeta_0)$ and *Student* (primary network) $f(.; h(\gamma))$ by approximating the similarity-dissimilarity between instances in a batch represented by the matrices $\mathcal{G}_m$ of a square matrix of batch-size. First, the feature maps are flattened to produce $Q^{f_m(x_i; h(\gamma))}, Q^{g_m(x_i; \zeta_0)} \in \mathbb{R}^{b \times |C_m|.w_m.h_m}$ by stacking the flattened feature maps of layer $m$ as rows .

$$Q^{f_m(x; h(\gamma))} = \text{reshape}(f_m(x; h(\gamma)); [b \times |C_m|.w_m.h_m]) ) \tag{14}$$

$$Q^{g_m(x; \zeta_0)} = \text{reshape}(g_m(x; \zeta_0)); [b \times |C_m|.w_m.h_m]) ) \tag{15}$$

Then the pairwise dot-product produces the $\mathcal{G}_{[i]}^{f_m(x_i; h(\gamma))}, \tilde{\mathcal{G}}^{g_m(x; \zeta_0)} \in \mathbb{R}^{b \times b}$ as Eq. 16. When each row $\tilde{\mathcal{G}}_{[i]}^{g_m(x; \zeta_0)}$ is normalised as Eq. 17, knowledge consisting of the similarity between instances in a batch is represented by matrices $\mathcal{G}^{f_m(x; h(\gamma))}, \mathcal{G}^{g_m(x; \zeta_0)} \in \mathbb{R}^{b \times b}$.

$$\tilde{\mathcal{G}}^{f_m(x; h(\gamma))} = (Q^{f_m(x; h(\gamma))})^\top . Q^{f_m(x; h(\gamma))}; \quad \tilde{\mathcal{G}}^{g_m(x; \zeta_0)} = (Q^{g_m(x; \zeta_0)})^\top . Q^{g_m(x_i; \zeta_0)} \tag{16}$$

$$\mathcal{G}_{[i]}^{f_m(x_i; h(\gamma))} = \frac{\tilde{\mathcal{G}}_{[i]}^{f_m(x_i; h(\gamma))}}{\|\tilde{\mathcal{G}}_{[i]}^{f_m(x_i; h(\gamma))}\|}; \quad \mathcal{G}_{[i]}^{g_m(x_i; \zeta_0)} = \frac{\tilde{\mathcal{G}}_{[i]}^{g_m(x_i; \zeta_0)}}{\|\tilde{\mathcal{G}}_{[i]}^{g_m(x_i; \zeta_0)}\|} \tag{17}$$

Similarity Preservation KD loss in Eq. 18 intends to approximate the primary network's (*Student*'s) similarity matrix between instances $\mathcal{G}^{f_m(x_i; h(\gamma))}$ to that of the pretrained *Teacher*'s $\mathcal{G}^{g_m(x_i; \zeta_0)}$ with equal network architecture, where $\|.\|_F$ represents the Frobenius norm.

$$\mathcal{L}_{SP-KD} = \sum_{m=1}^{M} \lambda_m . \|\mathcal{G}_{[i]}^{f_m(x_i; h(\gamma))} - \mathcal{G}_{[i]}^{g_m(x_i; \zeta_0)}\|_F^2 \tag{18}$$

### A.3.4 BACKGROUND ON JACOBIAN MATCHING (JACOBIANKD)

Jacobian Matching (termed in this paper as "JacobianKD") in Srinivas & Fleuret (2018) formulated the *Learning without Forgetting (LwF)* framework for knowledge distillation similar to Attention Transfer, implying the importance of matching the gradients between *Student* and *Teacher*, since the Jacobian represents the sensitivity-relationship between regions of the inputs with the regions of feature maps (or logits, originally). Since the full Jacobian matrix is computationally expensive, the gradient is utilised in practice. The mean squared error of L2-normalised gradients of (logit or) feature map w.r.t input $x_i$, conditioned by the class labels, represented by $\nabla_{x_i} f_m(x_i; h(\gamma))[y_i]$ and $\nabla_{x_i} g_m(x_i; \zeta_0)[y_i]$, are matched between primary (*Student*) $f$ and pretrained (*Teacher*) $g$ networks at each layer $m$ as in Eq. 19.

$$\mathcal{L}_{\text{JacobianKD}} = \sum_{m=1}^{M} \lambda_m . \sum_{(x_i, y_i) \in \mathcal{X}_B \times \mathcal{Y}_B} \left\| \frac{\nabla_{x_i} f_m(x_i; h(\gamma))[y_i]}{\|\nabla_{x_i} f_m(x_i; h(\gamma))[y_i]\|_2} - \frac{\nabla_{x_i} g_m(x_i; \zeta_0)[y_i]}{\|\nabla_{x_i} g_m(x_i; \zeta_0)[y_i]\|_2} \right\|_2^2 \tag{19}$$

### A.3.5 BACKGROUND ON CHANNEL-WISE KNOWLEDGE DISTILLATION (CWD)

Extending the spatial alignment of attention maps such as in AT Zagoruyko & Komodakis (2017), Channel-wise Distillation (CwD) (Shu et al. (2021)) aligns the feature maps of the *Student* (primary network) and the *Teacher* (pretrained network) in the channel dimensions. A probabilistic map is constructed as in Eq.20 where $f_m^c(x; h(\gamma))$ representing the primary network's feature map for only the channel $c$, and $\tau$ a temperature hyperparameter similar to the pioneering work on KD Hinton et al. (2015).

$$\phi(f_m^{c_m}(x; h(\gamma))) = \frac{exp(\frac{f_m^{c_m}(x; h(\gamma))}{\tau})}{\sum_{j=1}^{w_m.h_m} exp(\frac{f_m^{c_m}(x; h(\gamma))}{\tau})}; \quad \phi(g_m^{c_m}(x; \zeta_0)) = \frac{exp(\frac{g_m^{c_m}(x; \zeta_0)}{\tau})}{\sum_{j=1}^{w_m.h_m} exp(\frac{g_m^{c_m}(x; \zeta_0)}{\tau})} \tag{20}$$

The CwD knowledge distillation loss $\mathcal{L}_{CwD-KD}$ matches the two distributions of the *Teacher* (pretrained) and the *Student* (primary network) by summing the Kullback-Leibler (KL) Divergence between the calculated probabilistic distributions for each channel in Eq. 20.

$$\mathcal{L}_{CwD-KD} = \sum_{m=1}^{M} \lambda_m . \frac{\tau^2}{|C_m|} \sum_{c_m \in C_m} \sum_{j=1}^{w_m.h_m} \phi(g_m^{c_m}(x; \zeta_0)) . \log \left[ \frac{\phi(g_m^{c_m}(x; \zeta_0))}{\phi(f_m^{c_m}(x; h(\gamma)))} \right] \tag{21}$$

### A.3.6 BACKGROUND ON RELATIONAL KD (RKD)

Relational KD (RKD) (Park et al. (2019)) calculates the discrepancy between *Student* and *Teacher* by using Huber loss $l_\delta^{\text{Huber}}(.,.)$ where two types of input measures are used for the feature-level relationships between instances in a batch $\mathcal{X}_b$: (1) distances $\psi_D$ between two instances ( $f_m(x_i), f_m(x_j)$ ), and (2) the cosine $\psi_A$ of the angle $\Omega_{ijk}$ between triplets ( $f_m(x_i), f_m(x_j), f_m(x_j)$ ).

$$l_\delta^{\text{Huber}}(p, q) = \begin{cases} \frac{1}{2}(p-q)^2 & \text{if}|p-q| \leq 1 \\ |p-q| - \frac{1}{2} & \text{otherwise} \end{cases} \tag{22}$$

Distance between a Tuple based RKD (RKD-D)

Distance between two feature maps $\psi_D(f_m(x_i; h(\gamma)), f_m(x_j; h(\gamma)))$ as Eq.23 for the primary network(*Student*) and the pretrained (*Teacher*) network $\psi(g_m(x_i; \zeta_0), g_m(x_j; ))$ are compared for knowledge distillation as in Eq.25, where the normalisation term $\mu_m^b$ is the sum of pairwise-distances $\mathcal{X}_b^{\{.,.\}_2}$ as Eq. 24.

$$\psi_D(f_m(x_i), f_m(x_j)) = \frac{1}{\mu}\|f_m(x_i) - f_m(x_j)\|_2 \tag{23}$$

$$\mu_m^b = \frac{1}{|\mathcal{X}_b^{\{.,.\}_2}|} \sum_{(x_i, x_j) \in \mathcal{X}_b^{\{.,.\}_2}} \|f_m(x_i) - f_m(x_j)\|_2 \tag{24}$$

$$\mathcal{L}_{RKD-D} = \sum_{m=1}^{M} \lambda_m \cdot \sum_{(x_i, x_j) \in \mathcal{X}_b^{\{.,.\}_2}} l_\delta^{\text{Huber}}( \psi(f_m(x_i), f_m(x_j)), \psi(g_m(x_i), g_m(x_j))) \tag{25}$$

Angle between a Triplet based RKD (RKD-A)

For each triplet $(i, j, k)$ in a batch $\mathcal{X}_b$, the angle between vectors $(j \to i)$ and $(j \to k)$ as in Eq.26 is used as a metric of knowledge in RKD-A method which can be used for matching between the feature maps from pretrained (*Teacher*) network and that from the primary (*Student*) network as in Eq. 27. $\mathcal{X}_b^{()_3}$ represents triplets in batch $\mathcal{X}_b$.

$$\psi_A(f_m(x_i), f_m(x_j), f_m(x_k)) = \frac{1}{\mu} < \frac{f_m(x_i) - f_m(x_j)}{||f_m(x_i) - f_m(x_j)||}, \frac{f_m(x_k) - f_m(x_j)}{||f_m(x_k) - f_m(x_j)||} > \tag{26}$$

$$\mathcal{L}_{RKD-A} = \sum_{m=1}^{M} \lambda_m \cdot \sum_{(x_i, x_j, x_k) \in \mathcal{X}_b^{()_3}} l_\delta^{\text{Huber}}( \psi_A(f_m(x_i), f_m(x_j), f_m(x_k)), \psi_A(g_m(x_i), g_m(x_j), g_m(x_k))) \tag{27}$$

### A.3.7 BACKGROUND ON FLOW OF SOLUTION PROCEDURE (FSP)

The the direction of the flow between two layers in a single neural network by taking the inner product between feature maps from each channel of two layers. FSP Yim et al. (2017) uses matches two matrices $S^{f_m, f_{(m+1)}}$ for *Student* and $S^{g_m, g_{(m+1)}}$ are calculated as in Eq.28.

$$S_{[\alpha, \beta]}^{f_m, f_{(m+1)}} = \sum_{a=1}^{max(h_m, h_{(m+1)})} \sum_{b=1}^{max(w_m, w_{(m+1)})} \frac{(\delta_m^{sel} \cdot u_m) \circ f_m^\alpha(x; h(\gamma))[a, b] \times (\delta_{m+1}^{sel} \cdot u_{(m+1)}) \circ f_{(m+1)}^\alpha(x; h(\gamma))[a, b]}{max(h_m, h_{(m+1)}) \times max(w_m, w_{(m+1)})} \tag{28}$$

where $\alpha, \beta$ representing channels from layers $m, (m+1)$, respectively, and $a, b$ representing spatial coordinates of a feature map. Depending on which feature maps from layer $m$ or $(m+1)$ are larger, the projection using an adaptive average pooling 2D layer $u_m$ or $u_{(m+1)}$ gets used, represented by $\delta_{(.)}^{sel} = \arg \max_{m, m+1}(h_m \cdot w_m, h_{m+1} w_{m+1})$.

FSP loss term for a pair of layers $m$ and $m+1$ is defined as the descrepency between two matrices, $S^{f_m, f_{(m+1)}}$ for the *hypernetised* primary network and $S^{g_m, g_{(m+1)}}$ for the (pretrained) *Teacher* network. This is repeated for all *hypernetised* layers as pairs as Eq. 29.

$$\mathcal{L}_{FSP} = \frac{1}{|\mathcal{X}_B|} \sum_{x \in \mathcal{X}_B} \sum_{m=1}^{M-1} \lambda_m \left\| S^{f_m, f_{(m+1)}} - S^{g_m, g_{(m+1)}} \right\|_2^2 \tag{29}$$

## A.4 DYNAMIC WEIGHING OF $\lambda_m$ COEFFICIENTS FOR KNOWLEDGE DISTILLATION LOSS TERMS

When using hypernetworks for training multiple layers at different depths of a primary network, gradient flows to the hypernetwork in multiple pathways. Hence, by altering the coefficients for knowledge distillation loss $\lambda_m, m = 1, ..., M$, were altered based on epoch number. A gaussian spike moving from early to latter incrementally was used in this regard. Early epochs have high $\lambda_m$ for early layers. As training progresses, the latter layers receive high probability as Eq. 30.

$$\tilde{\lambda}_m^{(n)} = exp\left[ -\left( \frac{m - \frac{n}{N}(M-1)}{2\sigma^2} \right) \right] \tag{30}$$

Then each $\tilde{\lambda}_m^{(n)}$ term for layer $m$ is clipped with a threshold hyperparameter $\Lambda_{min}$ as $\hat{\lambda}_m^{(n)} = \max(\Lambda_{min}, \tilde{\lambda}_m^{(n)})$. Finally, each term is calculated by normalising it with $\lambda_m^{(n)} = \tilde{\lambda}_m^{(n)} / \sum_{m=0}^{M} \tilde{\lambda}_m^{(n)}$. In the experiments in section 4, the hyperparameters were set to $\lambda_0 = 0.1, \Lambda_{min} = 0.05, \sigma = 5.0$. This dynamic weight assignment was compared versus the uniform assignment of $\lambda_m^{(n)}, m > 0$ while keeping $\lambda_0$ at 0.1 for classification loss term.

## A.5 CONVERGENCE PLOTS FOR CIFAR100

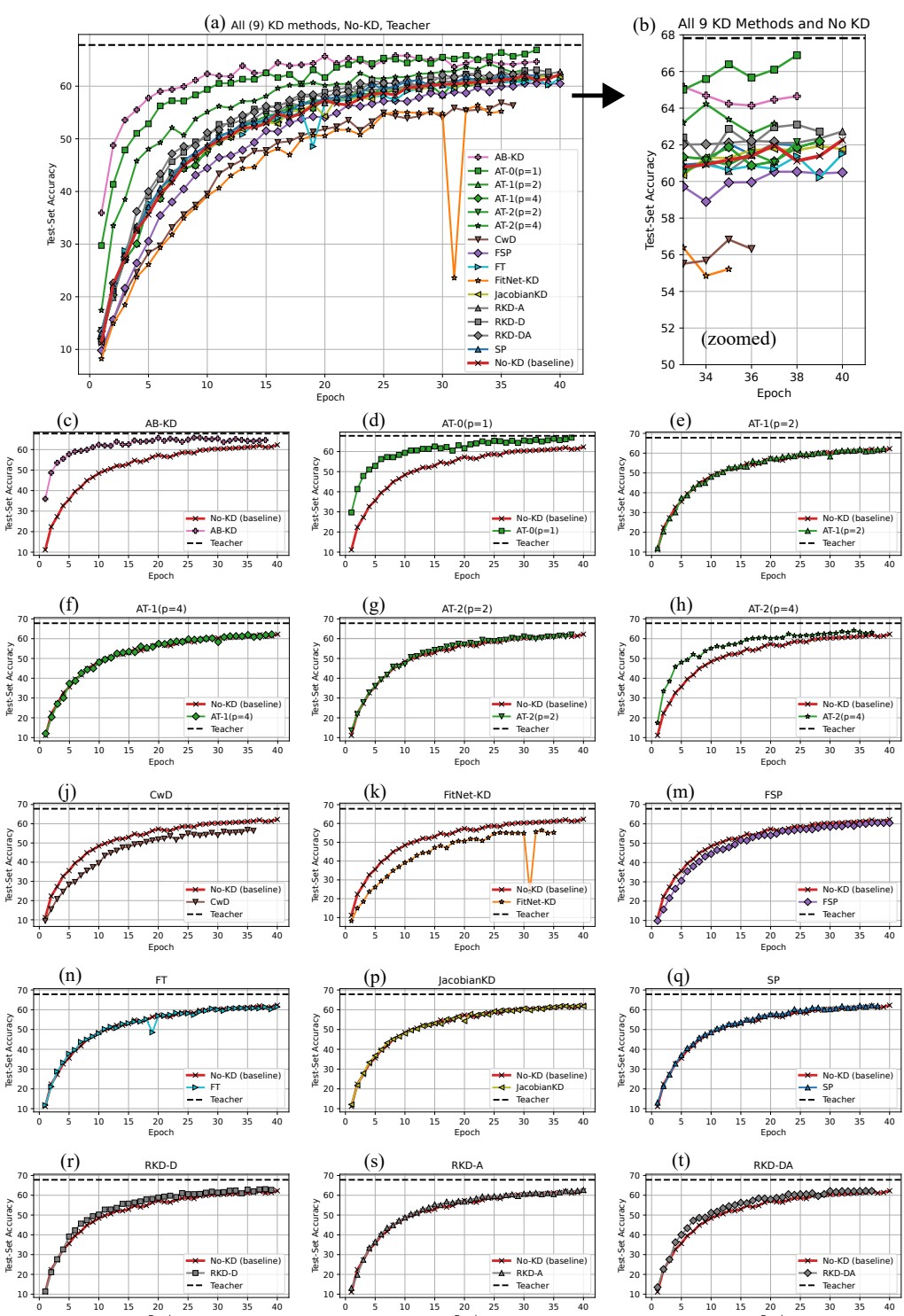

Figure 3: CIFAR100 results for Shufflenet_x0_5 as primary network with dynamic $\lambda_m$ (Section A.4) and pretrained *Teacher* network with nine knowledge distillation methods. (b) contains the zoomed patch from the last epochs 33-40. No-KD (baseline) is the MIP hypernetwork (Ortiz et al.)) without any knowledge distillation applied on the primary network. Each parameter configuration for each knowledge distillation method is indicated. For clarity, (a) and (b) are separately represented in (c) AB-KD, (d) AT-Type 0 (p=1), (e) AT-Type 1 (p=2), (f) AT-Type 1 (p=4), (g) AT-Type 2 (p=2) , (h) AT-Type 2 (p=4), (j) CwD, (k) FitNet-KD, (m) FSP, (n) FT, (p) JacobianKD, (q) SP, (r) RKD-D, (s) RKD-A, (t) RKD-DA together with *Teacher* and baseline (NoKD) are depicted. **Hypernetwork's hidden layer configuration is** [16, 64, 128]. Refer the column 2 of Table 1.

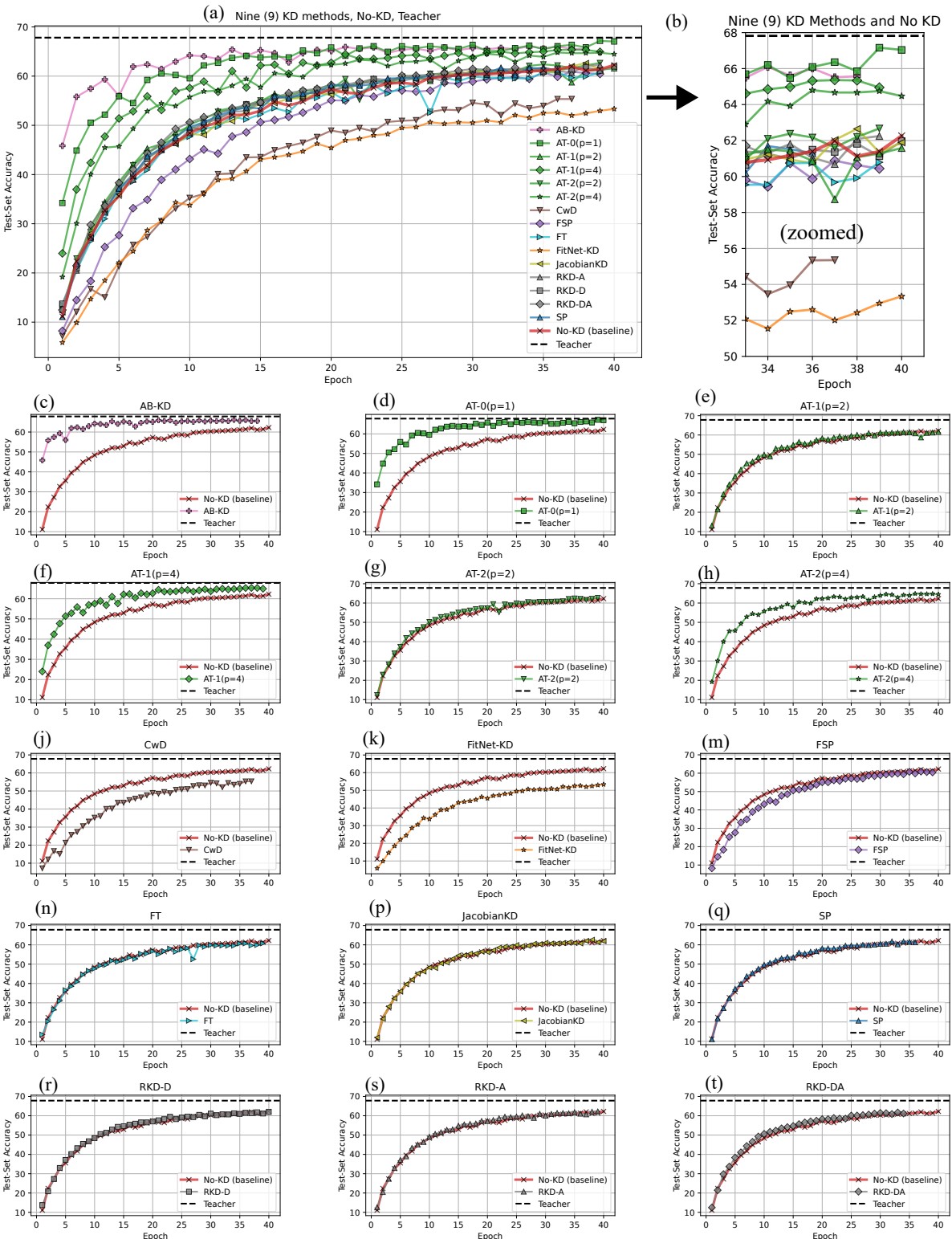

Figure 4: CIFAR100 test accuracy for Shufflenet_x0_5 with uniform $\lambda_m$ (Section A.4) as primary network and pretrained *Teacher* network with nine knowledge distillation methods. (b) contains the zoomed patch from the last epochs 33-40. No-KD (baseline) is the MIP hypernetwork (Ortiz et al.)) without any knowledge distillation applied on the primary network. Each parameter configuration for each knowledge distillation method is indicated. For clarity, (a) and (b) are separately represented in (c) AB-KD, (d) AT-Type 0 (p=1), (e) AT-Type 1 (p=2), (f) AT-Type 1 (p=4), (g) AT-Type 2 (p=2) , (h) AT-Type 2 (p=4), (j) CwD, (k) FitNet-KD, (m) FSP, (n) FT, (p) JacobianKD, (q) SP, (r) RKD-D, (s) RKD-A, (t) RKD-DA together with *Teacher* and baseline (NoKD) are depicted. **Hypernetwork's hidden layer configuration is** $[16, 64, 128]$. Refer the column 3 of Table 1.

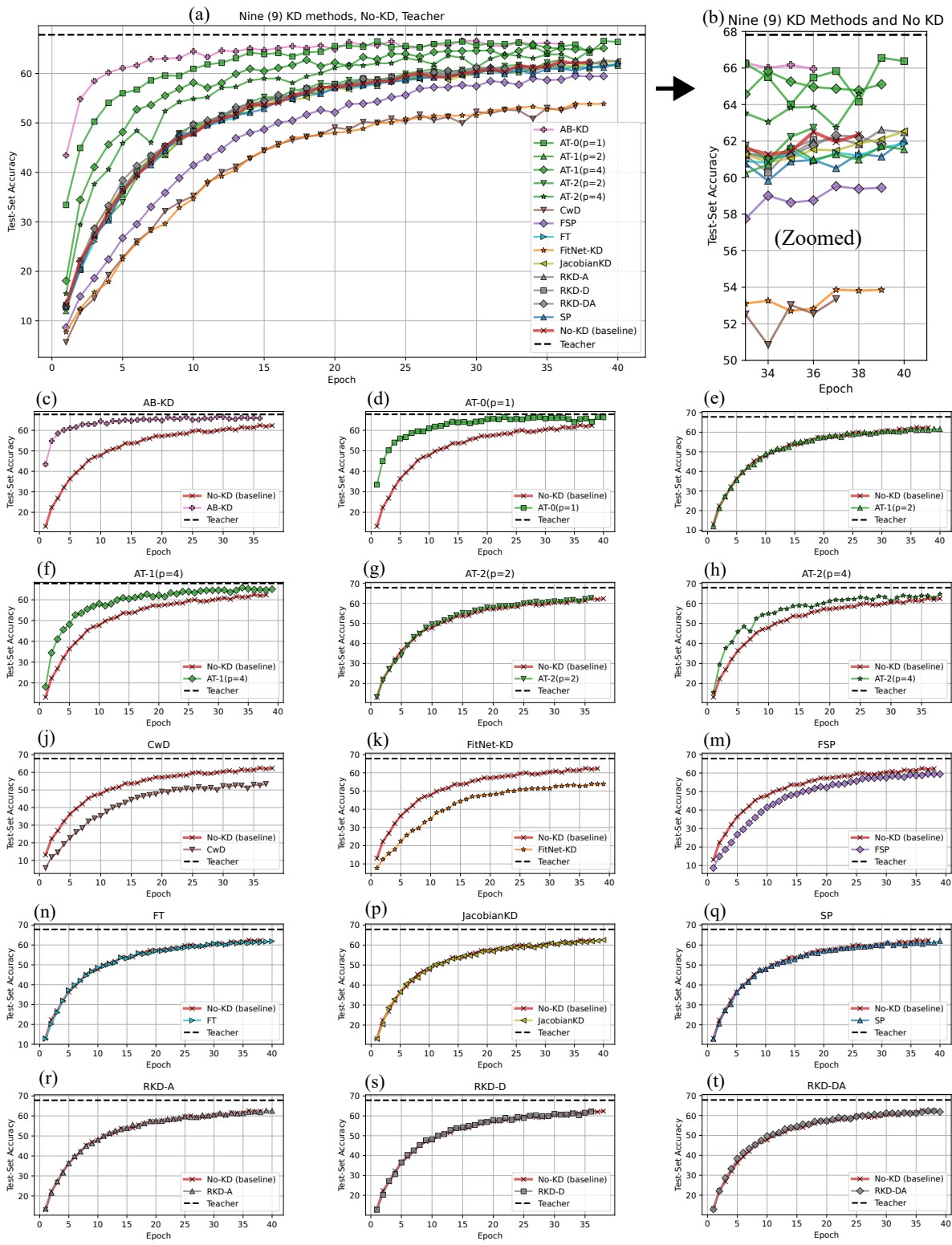

Figure 5: CIFAR100 test accuracy for `Shufflenet_x0_5` with uniform $\lambda_m$ (Section A.4) as primary network and pretrained *Teacher* network with nine knowledge distillation methods. (b) contains the zoomed patch from the last epochs 33-40. No-KD (baseline) is the MIP hypernetwork (Ortiz et al.)) without any knowledge distillation applied on the primary network. Each parameter configuration for each knowledge distillation method is indicated. For clarity, (a) and (b) are separately represented in (c) AB-KD, (d) AT-Type 0 (p=1), (e) AT-Type 1 (p=2), (f) AT-Type 1 (p=4), (g) AT-Type 2 (p=2) , (h) AT-Type 2 (p=4), (j) CwD, (k) FitNet-KD, (m) FSP, (n) FT, (p) JacobianKD, (q) SP, (r) RKD-D, (s) RKD-A, (t) RKD-DA together with *Teacher* and baseline (NoKD) are depicted. **Hypernetwork's hidden layer configuration is** $[16, 16, 16]$. Refer the column 4 of Table 1.

## A.6 Paraphraser Loss for Pre-Training in Factor Transfer

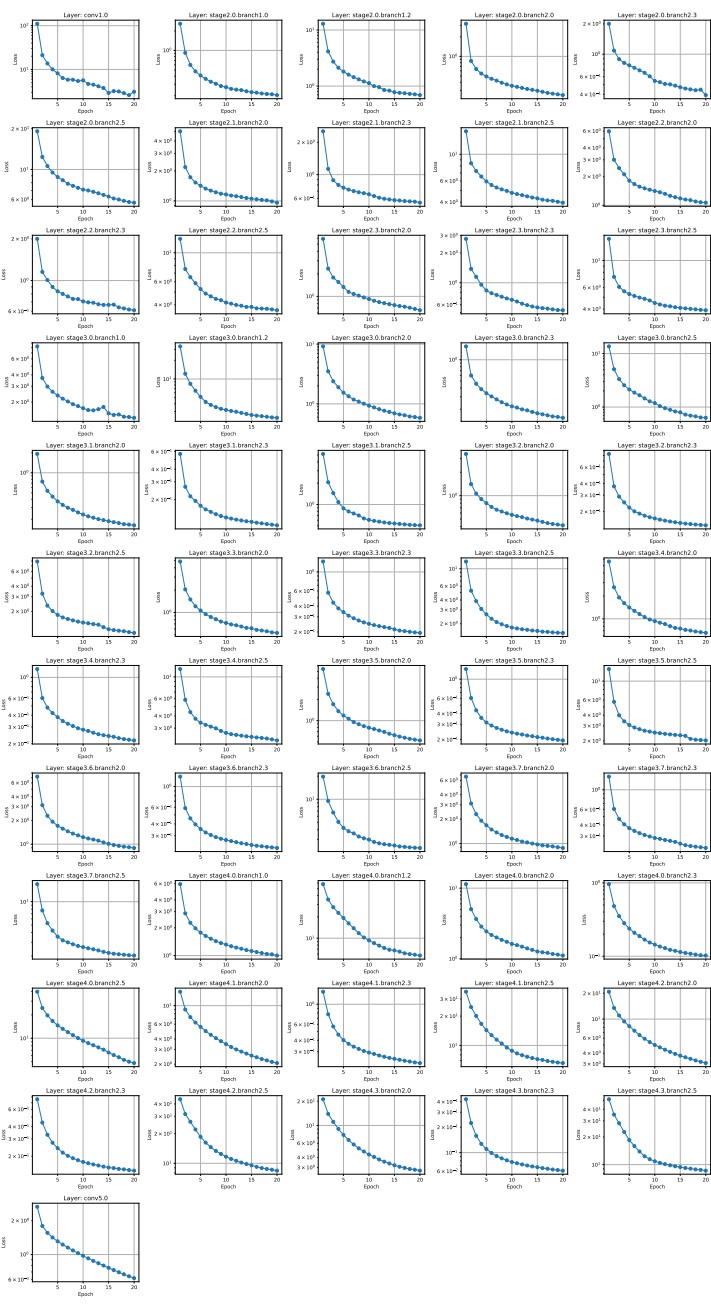

Figure 6: Reconstruction Loss of the *Paraphraser* as a pre-Training step in Factor Transfer prior to knowledge distillation in CIFAR100 classification. Details for fine-tuning these losses is provided in Section 3.4. Important to manually verify (only for this step) that the reconstruction loss of the *paraphraser* is within appropriate ranges as shown above.

## A.7   MAGNITUDE OF GRADIENT AT PENULTIMATE LAYER OF THE HYPERNETWORK FOR VGG19

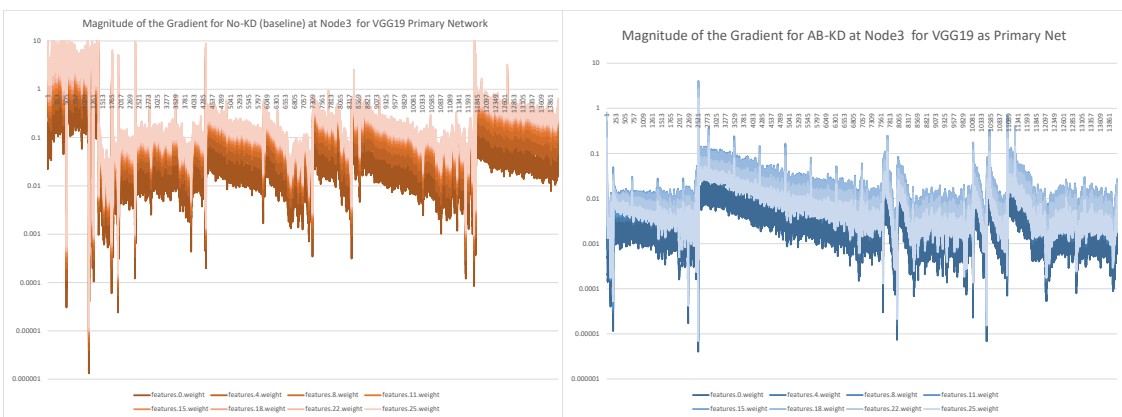

Figure 7: Magnitude of the gradient for a node (node 3) of the penultimate layer of the hypernetwork, indicating the gradient decay of the No-KD happening in VGG19 in the left figure. The right figure shows the reduction of the gradient magnitude for the same node when AB-KD is applied. Colours indicate early to last layers with increasing white-ness.

