# OpenReview forum: "Layer-wise Knowledge Distillation from a Pretrained Network Improves Hypernetwork Convergence"
_ICLR.cc/2026/Conference — Submitted to ICLR 2026_

### Official Review · Reviewer_s4Af · 2025-10-23

**Soundness:** 2
**Presentation:** 2
**Contribution:** 2
**Rating:** 2
**Confidence:** 5

**Summary:**

This paper investigates layer-wise knowledge distillation (KD) methods to improve hypernetwork convergence and test accuracy. Hypernetworks that generate weights for primary networks struggle with convergence and underperformance relative to canonically trained networks, especially when deeply hypernetized. The paper systematically evaluates 9 existing layer-wise KD methods (AB, AT, CwD, FitNets, FSP, FT, JacobianKD, RKD, SP) applied to the MIP (Magnitude Invariant Parameterisation) hypernetwork framework on CIFAR-100. Results show Attention Transfer (AT) and Activation Boundary (AB) KD significantly improve MIP, e.g., AB-KD achieves 72.65% accuracy on fully hypernetized VGG19 (vs. 11% baseline MIP, 73.87% canonically trained teacher), but most other methods fail or regress.

**Strengths:**

1. **Comprehensive empirical study**: Testing 9 KD methods systematically is thorough. Table 1 comparisons are informative for practitioners interested in hypernetworks.

2. **Dramatic performance improvement on pathological case**: VGG19 MIP improving from 11% → 72.65% with AB-KD is striking and demonstrates the method's value for extreme compression scenarios.

3. **Reproducibility**: Detailed hyperparameters, algorithms, and dataset specifications enable reproduction. Code release promised.

4. **Multiple architectures**: Testing on 10 different network types (ResNets, VGGs, RepVGGs, ShuffleNets, MobileNets) shows some generalization within CNN domain.

5. **Ablations on weighting strategy**: Comparing uniform vs. dynamic λ weighting (Table 1) provides practical guidance.

**Weaknesses:**

1. **Severely limited scope**:
   - **Single dataset (CIFAR-100)**: No ImageNet, STL-10, Tiny-ImageNet, or other benchmarks. CIFAR-100 is small; hypernetwork convergence dynamics may differ drastically on larger datasets.
   - **Niche problem**: Fully hypernetized deep networks is not a mainstream use case. MIP's original applications (Bayesian ConvNets, medical imaging) don't require full hypernetization.
   - **One setting only**: All experiments on image classification with CNNs. No other domains (NLP, graphs, etc.).

2. **Lack of mechanistic understanding**:
   - **Why do FitNets/CwD catastrophically fail?** FitNets regression of -7.04% is alarming but unexplained. Are they incompatible with hypernetwork gradients?
   - **Why is uniform λ better than dynamic λ?** Only hand-wavy "tug-of-war" explanation (line 402-404). No analysis, no ablation on Gaussian spike design.
   - **No analysis of what AT/AB capture** that other methods don't. Just empirical results.

3. **Incomplete experimental design**:
   - **No computational cost analysis**: KD adds 9 auxiliary losses. What's the training time overhead? Memory impact? This is critical for practitioners.
   - **Weak baseline comparison**: Only compares to MIP no-KD. What about other hypernetwork improvements (HyperFan, HyperInit)? Or simpler baselines like standard training with dropout/batch norm?
   - **No statistical significance testing**: Is 4.63% improvement significant? Standard deviations reported but no p-values.
   - **Incomplete ablations**: Why use 9 methods? Why not test on other KD methods (e.g., VID, RealKD, more recent approaches)?

4. **Mysterious design choices**:
   - Why [16,64,128] hidden dimensions for MIP? Why downscale to [16,16,16]? No justification or ablation.
   - Why use validation-based model selection for CIFAR-100? Standard is full training for N epochs.
   - Why only 40 epochs? VGG19 Figure 2 shows convergence is still occurring.
   - Factor Transfer requires 20 epochs of pretraining—no justification.

5. **Generalization concerns**:
   - **VGG19 exceptional case**: AB-KD achieves 72.65% (1.22% below teacher). But this is one datapoint. ResNet56 results are less impressive (3.62% gap).
   - **RepVGG paradox**: AB-KD *worse* than baseline on repvgg_a0/a2 (Table 2, -5.98%, -8.35%). Suggests method fragility. Why? No analysis.
   - **Smaller networks**: Results are mixed. AT/AB don't uniformly help all architectures.

6. **Questionable claims and framing**:
   - Abstract claims "first systematic study" of layer-wise KD for hypernetwork convergence, but the problem is so specialized this claim is somewhat trivial.
   - Paper positions this as solving a major problem, but hypernetwork convergence is not blocking progress in ML.
   - Title emphasizes "improves convergence" but mostly shows test accuracy improvement; convergence is only shown in plots.

7. **Missing practical guidance**:
   - When should practitioners use this method?
   - What's the memory/compute tradeoff?
   - Does it work beyond CIFAR-100?
   - How sensitive are results to hyperparameter choices (β, λ weights, etc.)?

**Questions:**

1. **Generalization**: Can you provide results on ImageNet or at least a larger dataset like STL-10 or Tiny-ImageNet? How different are conclusions?

2. **RepVGG failure**: Why does AB-KD fail dramatically on RepVGG architectures (-5.98%, -8.35%)? This suggests the method is architecture-dependent. How do you reconcile this with claims of broad applicability?

3. **Failure analysis**: FitNets/CwD regress 5-9%. Why? Are they incompatible with the implicit gradient flow in hypernetworks? Can you provide gradient analysis?

4. **Computational cost**: What is the wall-clock training time increase from adding layer-wise KD? Memory overhead? This is critical for practitioners deciding whether to adopt the method.

5. **Baseline methods**: Why not compare to other hypernetwork improvements (HyperFan, HyperInit)? Why not test other recent KD methods (VID, RealKD)?

6. **Statistical rigor**: Are the improvements (e.g., 4.63% for AT on ShuffleNet) statistically significant? Can you provide confidence intervals and significance tests?

7. **Dynamic λ design**: Why the specific Gaussian spike formula (Eq. 30)? Have you ablated the σ, Λ_min parameters? Why is uniform uniformly better?

8. **Scope justification**: What is the real-world use case for fully hypernetized deep networks on CIFAR-100? Can you motivate beyond compression?

---

> ### Author Response · Authors · 2025-12-03
>
> Thank you very much for your review and we really appreciate it.
>
> A. Choice of a single dataset – CIFAR-100:
> We used a single, moderately sized dataset (CIFAR-100), following prior works on hypernetwork convergence. This choice allows analysis of computationally and memory-intensive models and the layer-wise KD methods which are also complex in their design. In particular, the penultimate layer of the MIP hypernetwork makes the setup highly demanding: using hidden dimensions [16, 64, 128] increases memory consumption by up to 128× the total number of hypernetised parameters. We are currently working on experiments with TinyImageNet and will provide those results once ready.
>
> B. Choice of hidden dimensions of the hypernetwork:
> MIP (Ortiz et al., ICLR 2024) used hidden dimensions [16,16,16] and [16,64,128] in their experiments, so we adopted similar sizes. The penultimate layer was reduced from 128 to 16 to reduce computational and memory costs, as mentioned above.
>
> On your comment "Why [16,64,128] hidden dimensions for MIP? Why downscale to [16,16,16]? No justification or ablation.
> ", we provided the reasoning as quoted below in our paper:
> "Memory utilisation of MIP heavily depends on the number of nodes in its penultimate hidden layer (last hidden layer’s dimensionality), hence it had to be scaled from 128 to 16, so that the hidden layers had to be downscaled to [16, 16, 16] from [16, 64, 128] to facilitate execution on the server."
>
> C. Applications of this result:
> As stated in the paper:
> "Further, the proposed method that approximates a single Teacher favours one extreme of hypernetwork applications—improving a primary network on a single task or dataset, such as for pruning—over the other extreme of generalisation applications."
>
> D. Figure 2, VGG19 convergence: The validation set is used for model selection; a model is saved only if the validation accuracy improves. Across 100 epochs, all three methods stopped producing better models around the 95th epoch, as indicated by the absence of new markers in the plot. What is most important is the drastic difference between AB-KD and No-KD methods. Each epoch is extremely computationally and memory intensive, so we do not have the resources to run a larger number of epochs with our current facilities.
>
> E. Use of a validation set: Thank you for your comment. We require both a validation set and a separate test set, which is standard practice in machine learning. The validation set is used to prevent overtraining by guiding model selection, ensuring that the saved model is the one that generalises best.
>
> F. Statistical tests: Hypernetwork training, particularly with penultimate layer dimensions of 16 or 128 in the MIP hypernetwork, is extremely computationally and memory intensive. Each training run can take 4 hours to 2 days depending on the model complexity. Unfortunately, due to these constraints, we are unable to perform multiple runs for statistical significance testing, although we acknowledge its importance.
>
> G. Runtime information: We will provide a revised version including run-time details. We apologise for not including this information initially.
>
> H. Choice of 40 and 100 epochs for VGG models: Due to the extremely high computational cost of fully hypernetised MIP models, we relied on model selection using a validation set (saving only the best-performing model). The test accuracies in Figures 2–5 visually depict the convergence of the layer-wise KD methods. For VGG models, we used 100 epochs because shorter training did not allow convergence, given the additional computational burden on the primary network.
>
> Thank you again for your valuable feedback.

---

> ### Author Response · Authors · 2025-12-03
>
> Our focus in this work is on hypernetwork convergence. These results provide a foundation for future work on chunk-wise generation as a compression mechanism. We apologise for not explicitly mentioning this ongoing work, which clarifies its practical applicability. While we do not currently focus on generalisation across multiple tasks, an interested researcher could extend this approach using multiple teachers for broader KD applications. Addressing the unresolved problem of hypernetwork convergence is a key step toward enabling its practical use in compression tasks. In the broader context, we aim to apply this to edge-based bioacoustics for animal conservation, but the present work serves only as a stepping stone toward compression via chunk-wise generation, where we try to put both the hypernetwork and a chunk of the big network inside the edge device. This is also why we did not explore Transformers (larger models that are not used in edge based bioacoustics hardly) in this study. We cannot publicly broadcast these plans in detail, as it may compromise future publications.

---

### Official Review · Reviewer_MeM6 · 2025-10-30

**Soundness:** 4
**Presentation:** 4
**Contribution:** 3
**Rating:** 6
**Confidence:** 4

**Summary:**

This paper addresses the critical and well-known problem of convergence instability in hypernetworks, which generate the weights for a primary network. The authors observe that even recent advanced frameworks like Magnitude Invariant Parameterisation (MIP/HyperLight) fail catastrophically when applied to deep, fully hypernetised architectures without residual connections (e.g., VGG19 accuracy drops to 11%). To solve this, the paper proposes a unified framework, LayerKD-HN, which uses a pre-trained, architecturally identical "teacher" network to provide layer-wise supervision to the hypernetised "student" network via knowledge distillation (KD). The core hypothesis is that this provides shorter, more stable gradient paths and clearer optimization targets, mitigating the gradient conflicts ("tug-of-war") that destabilize training.The authors conduct a large-scale empirical study, systematically evaluating nine different layer-wise KD methods across ten diverse network architectures.The key findings are that Attention Transfer (AT) and Activation Boundary (AB) distillation are exceptionally effective, with AB-KD dramatically rescuing the VGG19 model's accuracy from 11.00% to 72.65%, bringing it within 1.22% of the canonically trained teacher network.

**Strengths:**

This work's main strength lies in its novel, well-motivated idea backed by extensive and compelling empirical evidence.
Novel and Impactful Problem-Solution Pairing: The paper creatively repurposes an existing family of techniques (layer-wise KD) to solve a different, highly relevant, and challenging problem (hypernetwork convergence). This is a significant conceptual contribution that moves beyond using KD for just compression or performance boosting, framing it as a crucial optimization tool.
Extremely Rigorous and Broad Empirical Evaluation: The systematic comparison of nine distinct KD methods across ten different CNN architectures is a standout feature.This comprehensive study provides a valuable service to the community by identifying which distillation strategies are most suitable for this challenging training regime. This level of rigor greatly increases the credibility and generality of the findings.
The performance improvements are not incremental but substantial and, in some cases, transformative. The ability to take a VGG19 model from a near-random 11% accuracy to 72.65% is a powerful demonstration of the method's efficacy. This result alone proves that the proposed framework can make previously intractable hypernetwork configurations viable.

**Weaknesses:**

Despite its strengths, the paper has several weaknesses that limit its overall impact and scientific depth.
Insufficient Contextualization and Missing Key Baselines: The paper almost exclusively benchmarks against the MIP/HyperLight framework without any KD.However, the field of hypernetwork convergence includes other important and orthogonal lines of work, most notably principled initialization methods like HyperFan and HyperInit.These methods tackle the exact same stability problem from a different angle (initialization vs. in-training regularization). A thorough comparison is necessary to understand if the proposed LayerKD-HN offers benefits beyond what can be achieved with better initialization alone, or if they are complementary. Without this comparison, the paper's claim to "improve hypernetwork convergence" is limited to the context of the MIP baseline, not the broader state-of-the-art.

Lack of Insight into Why Certain Methods Excel: The paper is an excellent report on what works (AT and AB) but provides very little analysis on why they are so effective while others (like FitNets) fail spectacularly.The most successful methods (AT, AB) transfer more abstract, structural knowledge (spatial attention, activation boundaries) rather than exact feature values.This suggests a deeper principle—that low-variance, simplified supervisory signals are crucial for stabilizing a chaotic training process—but this insight is not explored or articulated by the authors.

Unexplored Interaction Between KD Method and Architecture: Table 2 presents a fascinating result that is not discussed: AB-KD is the best method for VGG models (without residual connections) but performs worse than the baseline on RepVGG models (with residual connections), where AT-KD excels.This strongly suggests that the choice of the optimal KD method is architecture-dependent. This is a significant finding in its own right, but the paper misses the opportunity to analyze this interaction, which could have led to deeper insights about the interplay between explicit architectural stabilizers (residual connections) and external training stabilizers (KD).

**Questions:**

The paper's related work section mentions initialization-based convergence methods like HyperFan and HyperInit. Could the authors elaborate on why these were not included as experimental baselines? A direct comparison seems crucial for positioning the contribution of this work within the state-of-the-art. Furthermore, have the authors considered if their LayerKD-HN framework is complementary to these initialization schemes (e.g., applying LayerKD-HN on a HyperFan-initialized network)?

The empirical results clearly show that KD methods transferring abstract knowledge (AT, AB) are far superior to those matching raw feature values (FitNets).Do the authors have a more concrete hypothesis for this phenomenon? Could it be that for an unstable training dynamic like that of a hypernetwork, a "low-variance" or "simplified" KD signal is key, and that trying to match noisy, high-dimensional feature maps is counterproductive?

Table 2 shows a strong interaction effect: AB-KD is highly effective for VGG but detrimental for RepVGG, while AT-KD is consistently beneficial for both.What is the authors' interpretation of this? Could it be that the strong, binary constraints of AB-KD act as a necessary stabilizer for networks lacking residual connections, but become an overly restrictive prior for networks that already have stable gradient flow?

The proposed method requires an additional forward pass through a teacher network during training. Could the authors quantify the computational overhead (in terms of training time and memory) compared to the MIP baseline?

---

> ### Author Response · Authors · 2025-12-03
>
> Thank you very much for your extremely valuable review. Your comments reflect a deep understanding of the problem and even extend beyond what we have presented, offering insightful suggestions and potential pathways forward. It is clear that Reviewer MeM6 has thoroughly understood our work, and we are truly delighted to receive such constructive feedback, which will undoubtedly help us improve the paper.
>
> HyperFan and HyperInit, on Initialisation Techniques:
>
> We initially began experimenting with Hypernetworks from scratch and encountered several practical issues, including sudden training collapses caused by exploding values and other severe failures. Our observations were later reinforced by MIP (Ortiz et al., ICLR 2024), which emphasised in their related work section that existing approaches for improving hypernetwork convergence often fail in practice. In particular, the Parameter Initialization, Normalization Techniques, and Adaptive Optimization subsections highlight these challenges. As they state:
> “Our work demonstrates that existing initialization strategies can be ineffective when applied to hypernetworks.”
> But, we appreciate your comment on this and we will provide an experimental result to further analyse hypernetwork initialisation methods together with the LayerKD-HN, and as a baseline method. Thank you very much.
>
> We will provide further analysis on AB-KD and AT-0 methods while providing insights on why other methods fail. We agree with the reviewer on low-variance signal or a simplified KD signal being the key here. Furthermore, the AB-KD method being a noise-resistant approach stemming from SVM formulation is likely to result in a good convergence.
>
> Thank you very much for the feedback. Really appreciate it, and wonderful to have someone else understood our work very well.

---

### Official Review · Reviewer_gkqC · 2025-10-31

**Soundness:** 3
**Presentation:** 3
**Contribution:** 1
**Rating:** 2
**Confidence:** 4

**Summary:**

This work presents a study and evaluation of how layer-wise knowledge distillation from a fully converged neural network can improve the convergence of hypernetwork training. The study investigates nine (9) layer-wise distillation methods and their impact on the convergence of training for one hypernetwork architecture (HyperLite) trained on one dataset (CIFAR100).

A full set of all (9) layer-wise knowledge distillation is evaluated to generate a ShuffleNetv2.0x5 model, and a subset of two (2) layer-wise knowledge distillation is evaluated to generate ShuffleNet, ResNet, MobileNet, VGG, and Parameterised VGG models. Results indicate that layer-wise knowledge distillation does help with convergence and of hypernetwork training and performance of the generated neural network.

I very much appreciate the idea and think that hypernetworks are a great area to study. However, I have multiple concerns given the current state of this submission. My concerns are centred around the motivation of layer-wise knowledge distillation, the lack of methodological novelty, and the experimental setup. I will outline each of my concerns in the weakness section below.

**Strengths:**

- **(S1)**: I appreciate the work in the area of hypernetworks, as they have great potential to improve initialization or compression of larger neural networks
- **(S2)**: This paper is easy to read and follow, given its systematic build-up.

**Weaknesses:**

- **(W1)**: Motivation: I have difficulties understanding the utility of using layer-wise knowledge distillation using a fully converged teacher model. Do I understand correctly that the proposed method does require a fully converged model of the exact same neural network model (=architecture) trained on the exact same dataset as the teacher for distillation? Why would I do this? One would need to fully train a neural network to use the weights (or activations) as information for the training of a hypernetwork to generate the exact same neural network weights. The proposed method would make more sense when used in an out-of-distribution setup, where the teacher would have another architecture or would be trained on another dataset. In the current setup, it is difficult to see its utility.

- **(W2)**: Lack of novelty: This work provides an investigation of layer-wise knowledge distillation for hypernetwork training. The idea to use distillation is already seen in hypernetworks; the hypernetwork architecture used is known, and the layer-wise distillation methods are also known. There is minor methodological novelty in this setup. Regardless of this, the findings of the investigation are interesting, but looking at the findings, I have some concerns about the experimental setup.

- **(W3-1)**: Experimental setup: This work only evaluates one kind of hypernetwork (HyperLite) trained on one dataset (CIFAR100). To understand if the proposed method does generalize well, one would need to show its utility on multiple datasets and more than one kind of hypernetwork.

- **(W3-2)**: Experimental setup: Since knowledge distillation is known in the context of hypernetwork training, it would be important to see how "regular" knowledge distillation performs in comparison to layer-wise knowledge distillation. This baseline is missing.

**Questions:**

- **(Q1)**: Do you have any intuition about what would happen if you would set lambda_0 = 0, i.e., totally neglecting the "classification loss" of the generated neural network and only train with the distillation loss of the teacher network?

**Details Of Ethics Concerns:**

-

---

> ### Author Response · Authors · 2025-12-03
>
> (W1) Motivation
>
> Thank you for the question and for emphasising the need for a pretrained teacher with the same architecture. Yes, the proposed method (LayerKD-HN) investigates whether an already pretrained teacher can assist hypernetwork training when hypernetising an equivalent student network. This is a more challenging task because the hypernetwork generates weights for the primary network’s hypernetised parameters, which are placeholders; updates occur only in the hypernetwork. This implicit update scheme makes training unstable.
>
> Hypernetworks can later be used for chunk-wise generation (and pruning), which we are investigating as future work, and this work serves as a stepping stone. We did not emphasise this in the paper because it is ongoing work. Our result—that AB-KD and AT-0 help hypernetwork training—is therefore valuable.
>
> When we started, we found hypernetwork training extremely difficult due to exploding activations and collapses, as discussed in the MIP paper (Ortiz et al., ICLR 2024). Parameter initialisation, normalisation, and adaptive optimisers were found to be insufficient. We believe others who experience these difficulties will benefit from our finding that AB-KD and AT-Type0 can approximate the canonical model better (e.g., AB-KD with MIP achieved 72.65% vs. the teacher's 73.87%, compared to the 11% MIP baseline).
>
> (W2) Lack of novelty
>
> Thank you for raising this concern. Our focus is the practical challenge of hypernetwork convergence when fully hypernetising deeper networks. Full hypernetisation is required for our future work on chunk-wise generation (not emphasised in the paper; apologies). Although the methods used are existing ones, the contribution lies in improving hypernetwork convergence in a setting known to be extremely difficult, as highlighted in MIP. Implementing hypernetworks from scratch showed us how hard this is in practice. Our results provide useful guidance for future researchers, as also noted by the third reviewer (MeM6, below).
>
> (W3-1) Experimental setup: datasets and hypernetwork types
>
> Thank you for emphasising the need for more datasets. We are currently training ShuffleNetV2_0x5 on TinyImageNet. CIFAR100 was chosen because prior work on hypernetwork convergence used similar canonical datasets. Our main objective was to make a computationally expensive, fully hypernetised model such as VGG19 converge better.
>
> MIP was chosen because they are the most recent work presented in ICLR last year, with a clear emphasis on practical challenges and their findings on existing methods as indicated in the related work section of their paper. Please kindly refer to the section in Ortiz et al., ICLR 2024. It is not easy to test the 9 layerwise KD methods in a unified fashion. While appreciating your comments, our original intention was to avoid complicating the study with multiple older hypernetwork variants and KD methods. Instead, we focused on the core goal of improving hypernetwork convergence by providing shorter gradient pathways through layer-wise KD, supported by an already trained teacher model with an equivalent architecture. We believe this direction offers clearer insights and will be useful for the community.
>
> Hypernetwork hidden layer sizes—especially the penultimate layer—greatly increase memory usage. For example, MIP’s [16, 64, 128] hidden sizes scale memory by 128× the number of hypernetised parameters. This limits using larger images and more complex models. Our goal was first to unify nine layer-wise KD methods under LayerKD-HN and then identify which ones best improve convergence. TinyImageNet results will be presented when ready.
>
> (W3-2) Experimental setup: regular KD baseline
>
> Thank you for pointing this out. We will provide results for regular KD. However, based on our hypothesis and experience, hypernetwork training benefits from shorter gradient paths, which layer-wise KD provides. As Reviewer MeM6 noted, shorter and more stable optimisation paths help mitigate gradient conflicts. For this reason, we did not initially compare KD variants against each other; our emphasis was on solving hypernetwork convergence. Original KD does not offer shorter paths and may not support convergence. We used the MIP baseline because it is the most recent and comprehensive convergence work. Nonetheless, we will add regular KD results.
>
> (Q1)
>
> Thank you for the question. We will provide experiments with λ₀ = 0, completely removing the classification loss.

---

### Official Review · Reviewer_3QYY · 2025-11-01

**Soundness:** 3
**Presentation:** 2
**Contribution:** 2
**Rating:** 6
**Confidence:** 3

**Summary:**

The paper investigates hypernetwork convergence particularly for layer-wise knowledge-distillation setting. The paper discussed the issues with recent approaches and extended them with attention-transfer and activation-boundary methods. It performed extensive experiments across nine layer-wise knowledge distillation methods across nine different neural network architectures. The experiments demonstrated the superiority of proposed extensions, particularly for fully hypernised deep networks.

**Strengths:**

+ The paper investigates an important direction- both hypernetworks and knowledge distillation are becoming increasingly important in the current era of large models
+ The paper performs extensive experiments across different NN architectures and different KD methods
+ The experimental results supports the paper’s claims in the CIFAR 100 classification task

**Weaknesses:**

- The paper only explores convolutional networks and does not explored transformer-based models, whereas the most recent large models are mostly transformer-based. Furthermore, it is not clear if the advantages of LayerKD-HN will also hold for transformer-based models
- The presentation of the paper could be significantly improved. At its current form of the paper, too many concepts seems floating around and it is hard to keep track of the logic flow.
- Despite investing an important direction, the paper itself does not propose any significant novel method. It is more about combining things and see how they work in this case.
- The paper extends the MIP method and hence depends too much on the MIP paper. However, an ICLR paper should be self-contained.

**Questions:**

- How does the methods perform for transformer-based models?
- Why only CIFAR-100 classification task is used for benchmarking.

---

> ### Author Response · Authors · 2025-12-03
>
> Q1. Transformer-based models
>
> Thank you for highlighting the importance of transformer-based models for the presented research. Our approach does not rely on convolutional assumptions; LayerKD-HN only requires access to layer parameters for distillation, which also applies to transformer architectures. We have begun implementing a small-scale ViT experiment on CIFAR-100,and thank you very much for highlighting this. Appreciate it.
>
> A practical challenge arises with MIP-based hypernetworks applied to transformers: the final fully connected layer of the MIP hypernetwork grows proportionally to the size of the primary model. Using typical transformer with MIP hypernetwork's hidden sizes ([16, 64, 128], as in Gonzalez Ortiz, Guttag, and Dalca, 2024), produces a hypernetwork layer up to 128× larger than the corresponding transformer parameters, creating substantial memory and computational demands. Even a minimal configuration (16,16,16) still requires a 16× increase in memory. Our focus is not on any specific architecture, but on understanding how layer-wise knowledge distillation affects hypernetwork convergence for practically used deep networks. Given that the core problem is hypernetwork convergence and test-accuracy degradation, we avoided compounding it with the higher complexity of transformer architectures at this stage. When we first started using hypernetworks from scratch, we encountered the complexity of hypernetwork convergence which we believe the other researchers would encounter, and we believe our findings will benefit them, hence we reported these with ConvNets. Even the relatively old VGG models failed to converge when nearly fully hypernetised with MIP.
>
> Our method imposes no architectural restrictions. ConvNets were selected as a controlled setting with widely available pretrained models, enabling consistent comparison across nine KD methods and nine network families. Prior work, such as Magnitude-Invariant Parameterisation (MIP), also analysed convergence on ConvNets before extending to more complex architectures. This allows a clear and reproducible investigation of convergence behavior before moving to transformers.
>
> Q2. Dataset choice
>
> Thank you for emphasising the limitations of using only CIFAR-100. Based on your comments, we have begun experiments on TinyImageNet. The use of CIFAR-100 allowed us to evaluate nine heterogeneous knowledge distillation methods, each with different layer selections and student–teacher bridging strategies, in a unified and controlled setting. Datasets such as MNIST, CIFAR-10, and CIFAR-100 are widely used in hypernetwork convergence research for this reason. While larger datasets (e.g., ImageNet in HyperFan) exist, our primary goal was to address the convergence difficulties of computationally expensive and hard-to-hypernetise models such as VGG19. Once we achieved stable convergence and accuracy improvements, we shared these findings so that the community can further explore AB-KD and AT-Type0 for compression-oriented applications (the spectrum of hypernet applications; the presented method favours the single task + single dataset such as compression using chunk-wise generation or pruning using hypernetworks; This work we presented is a stepping stone to further development on hypernetwork applications with the LayerKD-HN).
>
> We agree that extending to additional datasets is valuable. We have begun implementation/experimentation to evaluate the strongest methods (AB-KD and AT-0) on TinyImageNet and on non-vision tasks such as ESC-50 for sound classification. Full hypernetisation incurs substantial memory and computational demands, so CIFAR-100 provides a practical, canonical setting for systematic evaluation across all nine layer-wise KD methods under LayerKD-HN. We will report TinyImageNet results once they are ready.

---

### Meta-Review · Area_Chair_y85E · 2026-01-07

**Summary:**

Major issues raised include:
1) Network diversity tested, only CNN models but not transformers

2) Write quality is concerned, with weak logic flow

3) Diversity of dataset evaluation is limited, only CIFAR100

4) Lack of novelty

5) Insufficient contextualization and connection with related works such as initialization methods e.g., HyperFan and HyperInit, and just limited to MIP/HyperLight framework.

6) Lack of insight into why some methods work better than the others

7) Missing analysis of running time

**Reviewer Concerns:**

In this response, the authors have covered the above issues as below

1) Not addressed: Agree to test a small ViT but no any progress updated, while discussed the limitations with transformers in this context in terms of the requirement of memory usage. It seems that this method is not generalisable to transformer in general.

2) Not addressed

3)  Not addressed: Agree to add a test on TinyImageNet but no progress yet

4) Further clarified the scope

5) Not addressed: Agree to add more analysis and experiments and no progress yet.

6) not addressed

7) not addressed

Overall this work is still preliminary, largely limited in many dimensions such as evaluation diversity, in-depth analysis, efficiency analysis, its position in the literature.

**Reviewer Scores:**

Given that the responses literarily address very little of concerns as raised by the reviewers, I do not see the hope of raising the scores.

---

### Decision · Program_Chairs · 2026-01-26

Reject